# The Ubiquitination of Arrestin3 within the Nucleus Triggers the Nuclear Export of Mdm2, Which, in Turn, Mediates the Ubiquitination of GRK2 in the Cytosol

**DOI:** 10.3390/ijms25179644

**Published:** 2024-09-06

**Authors:** Dooti Kundu, Xiao Min, Xiaohan Zhang, Xinru Tian, Shujie Wang, Kyeong-Man Kim

**Affiliations:** Department of Pharmacology, College of Pharmacy, Chonnam National University, Gwang-Ju 61186, Republic of Korea

**Keywords:** Mdm2, arrestin, GRK2, ubiquitination, Gβγ, clathrin

## Abstract

GRK2 and arrestin3, key players in the functional regulation of G protein-coupled receptors (GPCRs), are ubiquitinated by Mdm2, a nuclear protein. The agonist-induced increase in arrestin3 ubiquitination occurs in the nucleus, underscoring the crucial role of its nuclear translocation in this process. The ubiquitination of arrestin3 occurs in the nucleus, highlighting the pivotal role of its nuclear translocation in this process. In contrast, GRK2 cannot translocate into the nucleus; thus, facilitation of the cytosolic translocation of nuclear Mdm2 is required to ubiquitinate GRK2 in the cytosol. Among the explored cellular components and processes, arrestin, Gβγ, clathrin, and receptor phosphorylation were found to be required for the nuclear import of arrestin3, the ubiquitination of arrestin3 in the nucleus, nuclear export of Mdm2, and the ubiquitination of GRK2 in the cytosol. In conclusion, our findings demonstrate that agonist-induced ubiquitination of arrestin3 in the nucleus is interconnected with cytosolic GRK2 ubiquitination.

## 1. Introduction

Following agonistic stimulation of G protein-coupled receptors (GPCRs), Gα and Gβγ subunits of the associated heterotrimeric G protein dissociate [1]. The released Gβγ subunit, together with plasma membrane phospholipids, binds to the pleckstrin homology (PH) domain of GPCR kinase 2 (GRK2) located in the carboxyl terminus region. This binding recruits the inactive cytoplasmic GRK2 to the plasma membrane, where agonist-bound GPCRs are located [2,3,4]. GRK2 phosphorylates the receptors to provide high-affinity sites for arrestins [5,6]. Arrestins connect the activated receptors to endocytic adaptors, such as adaptor protein (AP)-2 and clathrin [7,8,9].

Both GRK2 and arrestin3 undergo Mdm2-mediated ubiquitination when receptors are activated by agonists [10,11,12]. The *Mdm2* gene, which was originally identified as an oncogene from transformed BALB/c mouse 3T3DM cells [13], functions as a RING finger-dependent E3 ubiquitin ligase [14,15,16]. Mdm2 contains a nuclear localization signal (NLS) and nuclear export signal (NES), and it constantly shuttles between the nucleus and the cytoplasm [17,18]. Mdm2 is present at much lower levels in the cytoplasm compared with the nucleus, and little Mdm2 is detected at the plasma membrane to regulate GPCRs [19]. Therefore, for a cytosolic protein to be ubiquitinated by Mdm2, either the cytosolic protein needs to move into the nucleus or Mdm2 must move out of the nucleus.

A previous study indicated that an increase in the Mdm2-mediated ubiquitination of arrestin3 occurs in the nucleus when GPCRs are activated by agonists and that the pivotal factor governing this process is the agonist-induced translocation of arrestin3 into the nucleus [12]. As expected from its main distribution in the cytosol, GRK2 does not possess an NLS [20]. Consequently, for GRK2 to undergo ubiquitination by Mdm2, Mdm2 needs to translocate to the cytoplasm. In our initial experiments, it was noted that upon activation of the dopamine D_2_ receptor (D_2_R), Mdm2 translocated from the nucleus to the cytosol. However, further studies are required to elucidate the mechanism underlying this phenomenon.

Despite some controversies, our experimental results suggest that Mdm2-mediated ubiquitination of arrestin3 increases in the nucleus, while GRK2 ubiquitination increases in the cytosolic region. Moreover, their ubiquitination processes are closely related. In this study, we investigated the molecular mechanisms involved in the mutual regulation of GRK2 and arrestin3 ubiquitination across different subcellular domains.

## 2. Results

### 2.1. Comparison of the Trafficking Patterns of Arrestin3 and GRK2 between the Nucleus and Cytosol

Mdm2, an E3 ubiquitin ligase, resides in the nucleus under basal conditions but dynamically translocates to the cytoplasm in response to external stimuli, such as GPCR agonists. This translocation leads to the ubiquitination of cytosolic proteins [21,22]. Thus, agonist-induced nuclear export of Mdm2 can be the critical cellular event in the Mdm2-mediated ubiquitination of cytosolic proteins.

Arrestin3—not GRK2—harbors the NLS, suggesting distinct interaction mechanisms with Mdm2. Interaction between arrestin3 and Mdm2 in the nucleus was reported after agonistic stimulation of GPCRs such as D_2_R, β_2_ adrenoceptor (β_2_AR), and angiotensin type 1 receptor [12]. In a previous study, agonistic stimulation of D_2_R led to the ubiquitination of arrestin3. However, arrestin3 with a mutated NLS did not undergo ubiquitination (Appendix A) [12]. The interaction between GRK2 and Mdm2 observed during agonistic stimulation of D_2_R was analyzed through subcellular fractionation, revealing that this interaction occurs in the cytosolic fraction (Figure 1A). These findings were further corroborated by fluorescence microscopy. As depicted in Figure 1B,C, arrestin3 and GRK2 were primarily localized in the cytosol under basal conditions. In cells exposed to leptomycin B (LMB), an inhibitor of nuclear export, a fraction of arrestin3 was detected within the nucleus, whereas GRK2 remained localized in the cytoplasm (comparing vehicle- and LMB-treated cells).

### 2.2. Arrestin, Gβγ, Clathrin, and Receptor Phosphorylation Are Involved in the Mdm2-Mediated Ubiquitination of GRK2

As a first step in understanding the molecular mechanisms involved in the nuclear export of Mdm2, the cellular components involved in the GRK2 ubiquitination were searched for. As reported previously [11], knockdown of arrestin3 abolished GRK2 ubiquitination (Figure 2A). The ubiquitination of GRK2, which was eliminated by arrestin knockdown, was reinstated by the co-expression of WT-arrestin3 but not by K11/12R-arrestin3 (Figure 2B). This lack of restoration by K11/12R-arrestin3 is due to a mutation in its Mdm2 binding site, preventing it from binding to Mdm2 [12,23]. These results suggest that arrestin3 is necessary for GRK2 ubiquitination and imply that the ubiquitination of arrestin3 is also crucial for this process.

Based on the results indicating that the ubiquitination of arrestin3 is implicated in the ubiquitination of GRK2 (Figure 2B), we hypothesized that the factors involved in the ubiquitination of arrestin3, for example, Gβγ and clathrin [12], would, likewise, play a role in the ubiquitination of GRK2.

As shown in Figure 3A,B, the sequestration of Gβγ or knockdown of clathrin heavy chain (CHC)—not caveolin1 (Cav1)—inhibited the ubiquitination of GRK2. Finally, a mutant of β_2_AR, GRK2-KO-β_2_AR, in which GRK2-mediated phosphorylation sites (T360, S364, S396, S401, S407, and S411) were altered, failed to mediate GRK2 ubiquitination (Figure 3C), suggesting that receptor phosphorylation is also involved in the ubiquitination of GRK2.

Based on these results, we next investigated how these cellular components regulate the nuclear export of Mdm2, which is a critical process necessary for the ubiquitination of GRK2.

### 2.3. Arrestin3 Shuttles between the Nucleus and Cytosol, Controlling the Subcellular Localization of Mdm2 and the Ubiquitination of GRK2

Because the translocation of Mdm2 from the nucleus to the cytosol is necessary for the ubiquitination of cytosolic proteins, we examined whether the cellular components shown in Figure 3 are involved in the nuclear export of Mdm2.

Initially, we investigated the impact of arrestins on the subcellular trafficking of Mdm2. As depicted in Figure 4A, under basal conditions, Mdm2 was primarily localized in the nucleus in cells expressing D_2_R, regardless of arrestin knockdown (Veh-treated groups). However, upon dopamine (DA) treatment, Mdm2 translocated from the nucleus to the cytosolic region (Figure 4A, DA-treated/Con-KD). Notably, this translocation was impeded by cellular arrestin knockdown (DA-treated/Arr-KD).

Because the agonist-induced cytosolic translocation of Mdm2 occurs in an arrestin-dependent manner, we were curious as to whether the trafficking properties of arrestin3 are associated with its ability to facilitate the agonist-induced nuclear export of Mdm2. For this, we utilized the mutants of arrestin3 in its nuclear localization signal (NLSX) [24] and nuclear export signal (NESX, L395A) [25]. In NLSX-arrestin3, seven amino acid residues (158–162 and 170–171) were mutated to alanine.

As illustrated in Figure 4B,C, co-expression of WT-arrestin3 but not NLSX- or NESX-arrestin3 restored GRK2 ubiquitination. These findings, along with the results shown in Figure 2B, indicate that the nuclear entry of arrestin3, its interaction with Mdm2 in the nucleus, and the subsequent nuclear export of arrestin3 are crucial for the D_2_R activation-mediated nuclear export of Mdm2, which is necessary for GRK2 ubiquitination.

This assumption was further supported by two experiments. First, GRK2 ubiquitination was inhibited when cells were treated with LMB (Figure 5A), which inhibits exportin 1 and blocks the nuclear export of proteins [26]. Next, we examined the nucleocytoplasmic shuttling of arrestin3 and Mdm2 (Figure 5B). Approximately 1 min after DA treatment, arrestin3 translocated from the cytosol to the nucleus, then returned to the cytoplasm around 5 min post treatment. Mdm2 began relocating from the nucleus to the cytoplasm about 3 min after DA treatment.

### 2.4. Gβγ and Clathrin Regulate the Nuclear Shuttling of Arrestin3, Which Determines the Nuclear Export of Mdm2

Gβγ and clathrin play roles in the ubiquitination of GRK2 (Figure 3A,B), while arrestins are particularly crucial, as they shuttle between the cytoplasm and nucleus [12]. Clathrin constitutes a major component of coat proteins found on clathrin-mediated endocytic vesicles [27,28]. To clarify their roles in the nuclear shuttling of Mdm2 and arrestin3, we investigated the effects of Gβγ sequestration and CHC knockdown on the nucleocytoplasmic shuttling of Mdm2 and arrestin3.

Under basal conditions, Mdm2 was predominantly localized in the nucleus (Figure 6A, Veh/Mock). Agonistic activation of D_2_R caused Mdm2 to relocate from the nucleus to the cytoplasm (DA/Mock). However, co-expression of GRK2-CT (GRK2 carboxyl terminus), which sequesters Gβγ [29], prevented the agonist-induced cytoplasmic translocation of Mdm2. As depicted in Figure 6B in the Mock/CHC-KD panel, knocking down CHC did not alter the subcellular distribution of Mdm2 and D_2_R, but it did prevent the DA-induced nuclear export of Mdm2.

In its basal state, arrestin3 was primarily located in the cytosol (Figure 6C, Veh/Mock). However, upon DA stimulation following LMB pretreatment (LMB∙DA/Mock), a portion of arrestin3 translocated to the nucleus. This translocation was partially inhibited by sequestering Gβγ (LMB∙DA/GRK2-CT) and more extensively hindered by CHC knockdown (LMB∙DA/CHC-KD).

These findings suggest that Gβγ and clathrin are essential for the nuclear entry of arrestin3, which, consequently, influences the nuclear export of Mdm2.

### 2.5. Receptor Phosphorylation Mediate Arrestin3 Ubiquitination and Nuclear Export of Mdm2, Which Are Needed for GRK2 Ubiquitination

Next, we determined the roles of receptor phosphorylation in the ubiquitination of GRK2. For this, GRK2-KO-β_2_AR, in which the consensus GRK2 phosphorylation sites are mutated [30], and D_2_R-IC23 with all serine and threonine residues in the second and third intracellular loops mutated [31] were utilized.

As shown in Figure 7A, GRK2-KO-β_2_AR, which failed to mediate GRK2 ubiquitination (Figure 3C), also could not mediate arrestin3 ubiquitination. Agonistic stimulation of D_2_R caused the nuclear export of Mdm2 (Figure 7B, comparing Veh- and DA-treated cells expressing WT-D_2_R). In contrast, D_2_R-IC23 failed to mediate the nuclear export of Mdm2 (cells expressing D_2_R-IC23). These results suggest that receptor phosphorylation is necessary for the ubiquitination of arrestin3 and the nuclear export of Mdm2, both of which are required for GRK2 ubiquitination.

### 2.6. Importin Complex Is Involved in the Nuclear Import of Mdm2 and Arrestin3

The nuclear movement of cytosolic protein molecules is commonly explained by the classical nuclear import pathway that employs the importin α/β complex [32,33]. In this pathway, a protein with a classical basic NLS, serving as the cargo, binds to adaptor protein importin α. Together with importin β, which attaches to the cargo and facilitates interactions with the nuclear pore complex, the cargo is transported to Ran GTP in the nucleus [34,35]. Mdm2 possesses a clear classical NLS [17].

Arrestin3, which was located in the cytosol (Figure 8A, left/upper panel), was observed both in the cytosol and nucleus when cells were treated with LMB (Figure 8A, left/lower panel). When cellular importin β1 was knocked down, arrestin3 largely failed to translocate to the nucleus (Figure 8A, right panels).

Mdm2, which was located within the nucleus in the basal state (Figure 8B, Veh/Con-KD), was located in both the nucleus and cytosol after DA treatment (DA/Con-KD). Upon knockdown of cellular importin β1, the localization of Mdm2 was detected in both the cytosol and nucleus in the basal state (Figure 8B, Veh/Impo-KD cells), suggesting that a portion of Mdm2’s nuclear entry under basal conditions was compromised. Because Mdm2 was already situated in the cytoplasm under basal conditions in importin β1-KD cells, it was challenging to conclusively ascertain whether nuclear export was further augmented by DA treatment.

Subsequently, we investigated whether the nucleo-cytoplasmic shuttling of arrestin3 and Mdm2 mediated by the importin1 complex is implicated in GRK2 ubiquitination in the cytoplasm. As depicted in Figure 8C, activation of D_2_R resulted in GRK2 ubiquitination, which was hindered by importin β1 knockdown.

These findings highlight that the nuclear entry of arrestin3 and Mdm2 facilitated by the importin1 complex, along with the ubiquitination of arrestin3 within the nucleus via this mechanism, is a pivotal process for GRK2 ubiquitination in the cytoplasm.

### 2.7. Interaction between Importin β1 and Arrestin3 Is Supported by Gβγ, Clathrin, and Receptor Phosphorylation

According to our findings, the ubiquitination of GRK2 relies on the cytosolic translocation of Mdm2, which, in turn, is contingent upon the interaction between arrestin3 and Mdm2 within the nucleus. This process is facilitated by the nuclear entry of arrestin3 through the importin complex. As Gβγ, clathrin, and receptor phosphorylation are pivotal for GRK2 ubiquitination, we investigated their potential involvement in the final phase of the cascade responsible for GRK2 ubiquitination—specifically, the nuclear entry of arrestin3 facilitated by the importin complex.

As anticipated, the interaction between importin β1 and arrestin3 increased upon agonist stimulation of D_2_R. However, this augmented interaction was nullified by sequestering Gβγ (Figure 9A). Additionally, the knockdown of clathrin but not caveolin1 also hindered the interaction (Figure 9B). Finally, the impact of receptor phosphorylation on their interaction was assessed using D_2_R-IC23. The augmented interaction between importin β1 and arrestin3 triggered by agonist stimulation of D_2_R was not observed with D_2_R-IC23, a mutant lacking phosphorylation sites [31] (Figure 9C).

In summary, these findings highlight that the ubiquitination of GRK2 by Mdm2 is regulated by the translocation of arrestin3 between the nucleus and cytosol, along with its interaction with Mdm2 within the nucleus. Gβγ, clathrin, and receptor phosphorylation play crucial roles in facilitating these processes.

## 3. Discussion

GRK2 and arrestins play crucial roles in the regulation and signaling of GPCRs [5,7]. In addition, both GRK2 and arrestin3 are targets of ubiquitination by Mdm2, where arrestin3 ubiquitination plays a crucial role in regulatory processes such as arrestin3 downregulation and receptor endocytosis [10]. GRK2 is also ubiquitinated by Mdm2, leading to its downregulation [11].

There are two ways for arrestin3 and GRK2, which reside in the cytosol, to encounter Mdm2, which primarily exists in the nucleus. Either arrestin3 and GRK2 enter the nucleus or Mdm2 moves to the cytosol. Arrestin3, possessing an NLS, moves into the nucleus upon receptor agonist stimulation and is ubiquitinated by Mdm2 [12]. In contrast, GRK2 lacks an NLS [20] and, thus, requires Mdm2 to relocate from the nucleus to the cytosol, a process dependent on the ubiquitination of arrestin3 in the nucleus (Figure 2). Consequently, this study focused on the regulatory pathways associated with arrestin3 ubiquitination, which is closely linked to the nucleocytoplasmic translocation of Mdm2.

The overall process of GRK2 ubiquitination triggered by agonist stimulation of GPCRs is depicted in Figure 10. Fundamentally, the critical step determining cytosolic GRK2 ubiquitination appears to be step 5, involving the movement of Mdm2 from the nucleus to the cytoplasm.

According to our experimental results, arrestin (Figure 2A)—in particular, its ubiquitination (Figure 2B)—is required for the ubiquitination of GRK2. Hence, it can be inferred that the factors governing the nuclear entry of arrestin3 (steps 1 and 2), crucial for its ubiquitination, likewise influence the ubiquitination of GRK2. This hypothesis was, indeed, confirmed by our study findings (Figure 5 and Figure 8). As Mdm2 is a nuclear protein and both arrestin3 and Mdm2 contain NES, it is presumed that Mdm2 and ubiquitinated arrestin 3 relocate from the nucleus to the cytoplasm, driven by the concentration gradient or with the help of exportin powered by the energy of GTP [36,37] (step 5, Figure 5B).

Reportedly, clathrin is required for GRK2-mediated receptor phosphorylation [38]. This function of clathrin was difficult to anticipate due to its typical role in facilitating receptor endocytosis [39,40]. However, it can now be elucidated by clathrin’s participation in the ubiquitination of GRK2 (Figure 3B). This is one piece of evidence supporting the idea that clathrin has functions beyond simply being a coated protein of endocytic vesicles.

According to previous reports, the cytosolic localization of Mdm2 can result from the overexpression of arrestin3 [41] or the presence of GPCRs with a high affinity for Gβγ [42,43]. This phenomenon is significant, as it may relate to GPCR desensitization characteristics [44,45], but it is observed in the basal state without GPCR activation. In contrast, the increase in nuclear ubiquitination of arrestin3 discussed in this study occurs in a dynamic situation induced by agonistic stimulation of GPCRs.

Arrestin-3 has previously been shown to be ubiquitinated when bound to a GPCR on the plasma membrane [10]. While this appears to conflict with our results, the discrepancy lies in the timing of arrestin3 ubiquitination. The earlier study did not consider the interaction between arrestin3 and Mdm2 in the nucleus, leading the authors to observe ubiquitination at the cytosolic interface of the plasma membrane.

Many receptors, including GPCRs, undergo simultaneous signaling and regulation processes when stimulated by agonists. Strong amplification at each step of receptor signaling is crucial for the efficient operation of the signaling system. GRK2 and arrestin3 play key roles in the regulation of GPCRs. Similar to the signaling process, the regulation process likely requires amplification at each step for efficient functional performance. However, there is a lack of mechanistic understanding of how this regulatory process is amplified. In this context, we believe that the positive correlation between the interdependence of GRK2 and arrestin3 ubiquitination and receptor phosphorylation observed in this study provides an important perspective for understanding the mechanism of GPCR regulation.

## 4. Materials and Methods

### 4.1. Materials

Leptomycin B (LMB), dopamine (DA), isoproterenol (ISO), MβCD, 4′,6-diamidine-2′-phenylindole (DAPI), methyl-β-Cyclodextrin (MβCD), rabbit FLAG antibodies (AB_439687), monoclonal antibodies against β-actin (AB_476743), agarose beads coated with anti-M2 FLAG antibodies, and goat anti-mouse or anti-rabbit (AB_258426 and AB_257896) horse radish peroxidase (HRP)-labeled secondary antibodies were obtained from Sigma-Aldrich Chemical Co. (St. Louis, MO, USA). Mouse monoclonal antibodies against clathrin heavy chain (CHC) (AB_397865) and caveolin 1 (Cav1) (AB_397470) were purchased from BD Biosciences (San Jose, CA, USA). Monoclonal antibodies against hemagglutinin (HA) (AB_783677), Mdm2 (AB_627920), and importin β1 (AB_2133993) were purchased from Santa Cruz Biotechnology (Dallas, TX, USA). Rabbit antibodies against arrestin 2/3 (AB_10547883) were purchased from Cell Signaling Technology (Danvers, MA, USA). Goat Alexa 594-conjugated anti-rabbit antibodies (AB_2534079) and Alexa 647-conjugated anti-rabbit antibodies (AB_2535813) were purchased from Thermo Fisher Scientific (Waltham, MA, USA). [^3^H]-Methylspiperone (84.2 Ci/mmol) and [^3^H]-CGP-12177 (41.7 Ci/mmol) were purchased from PerkinElmer Life Sciences (Waltham, MA, USA).

### 4.2. DNA Constructs

HA-Ub, FLAG-arrestin3, arrestin3-GFP, K11/12R-arrestin3-GFP, NLSX-arrestin3-GFP, NESX (L395A)-arrestin3-GFP, GRK2, FLAG-GRK2, GRK2-GFP, FLAG-Mdm2, Mdm2-GFP, GRK2-CT, D_2_R, FLAG-D_2_R, D_2_R-IC23, β_2_AR, and GRK2-KO-β_2_AR were described previously [12,28,30,31,46]. In GRK2-KO-β_2_AR, the consensus phosphorylation sites for GRK2 (T360, S364, S396, S401, S407, and S411) were mutated [30]. Since the serine and threonine residues in the second and third intracellular loops of D_2_R-IC23 were mutated, this construct was utilized to study phosphorylation-independent receptor functions [31,47].

### 4.3. Cell Culture

Human embryonic kidney (HEK-293) cells were obtained from the American Type Culture Collection (Manassas, VA, USA) and cultured in minimal essential medium supplemented with 10% fetal bovine serum, 100 units/mL penicillin, and 100 µg/mL streptomycin (Thermo Fisher Scientific) in a humidified atmosphere at 5% CO_2_. Polyethyleneimine (MW 25,000; Polysciences, Inc., Warrington, PA, USA) was used for transfection. Because PEI is a cationic polymer, it forms a complex with negatively charged DNA, facilitating its entry into cells. Mdm2-knockdown (KD) cells, CHC-KD cells, Cav1-KD cells, and importin β1-KD cells were prepared by a selection of the cells stably transfected with shRNA in pLKO.1 using 1 μg/mL puromycin [21,28,48,49]. Arrestin2/3-KD cells were prepared by simultaneous transfection of shRNAs targeted against arrestin2 and arrestin3, which was purchased from Promega (Madison, WI, USA).

### 4.4. Immunoprecipitation and Immunoblotting

For immunoprecipitation, target proteins tagged with FLAG epitope were expressed in HEK-293 cells. The cells were lysed using either RIPA buffer (150 mM NaCl, 50 mM Tris pH 8.0, 1% NP-40, 0.5% deoxycholate, and 0.1% SDS) or glycerol lysis buffer (10 mM HEPES, pH 7.4, 150 mM NaCl, 10% glycerol, and 0.1% NP-40) for 1 h at 4 °C. Subsequently, the cell lysates were centrifuged for 30 min at 14,000× *g*. The resulting supernatants were combined with 25 μL of anti-FLAG agarose beads (50% slurry) and incubated for 2–3 h at 4 °C. The beads underwent three washes with ice-cold washing buffer (50 mM Tris pH 7.4, 137 mM NaCl, 10% glycerol, and 1% NP-40), each lasting 5 min.

The immunoprecipitates were analyzed on SDS-PAGE gels and transferred to nitrocellulose membranes (Thermo Fisher Scientific). The membranes were incubated overnight at 4 °C with primary antibodies against the target proteins, followed by a 1 h incubation at 20 °C with HRP-conjugated secondary antibodies. Protein bands were visualized using a chemiluminescent HRP substrate (Thermo Fisher Scientific). Target proteins were detected with a chemiluminescent substrate, and the immunoblots were quantified using the ChemiDoc MP imaging system (BioRad, Hercules, CA, USA).

### 4.5. Immunocytochemistry

Cells expressing the target proteins were seeded onto coverslips and fixed with a solution of 4% paraformaldehyde and 0.2% Triton X-100 in PBS for 20 min at 20 °C. They were then incubated for 1 h with PBS containing 3% FBS and 1% bovine serum albumin, followed by a 1 h incubation with antibodies specific to the target proteins at 20 °C. After three washes, the cells were incubated with Alexa 595-conjugated anti-rabbit or anti-mouse antibodies at a 1:500 dilution. Following another three washes, the cells were mounted on slides using Vectashield (Vector Laboratories, Burlingame, CA, USA) and visualized with a laser scanning confocal microscope (TCS SP5/AOBS/Tandem, Leica Microsystems GmbH, Wetzlar, Germany). The immunocytochemistry experiment was performed 2–3 times, and a total of 5 to 10 cells per experimental group were analyzed for each experiment.

### 4.6. Radioligand Binding

HEK-293 cells transfected with D_2_R were seeded in 24-well plates. The following day, the cells were rinsed with 0.5 mL of prewarmed serum-free medium. Subsequently, the cells were incubated at 4 °C for 150 min with 250 µL of [^3^H]-methylspiperone (final concentration, 1 nM) in the absence or presence of a competitive inhibitor (10 µM haloperidol). After incubation, the cells were washed three times with the same medium, and 1% SDS was added. The samples were then mixed with scintillation fluid and counted using a liquid scintillation analyzer (Perkin Elmer, 1450 MicroBeta TriLux, Waltham, MA, USA).

### 4.7. Image Processing

The images were imported into the Fiji version of ImageJ for processing. To assess the spatial overlap among different fluorescent labels (co-localization), the Pearson correlation coefficient (γ value) was used [50]. Co-localization levels were then classified into the following three categories: strong (γ value from 0.50 to 1), moderate (0.30 to 0.49), and weak (below 0.29). The circumference of the cell was measured using automated software (NIS-Elements AT program, version AR 4.50; Nikon Inc.) (Nikon Inc., Tokyo, Japan).

### 4.8. Detection of Protein Ubiquitination

HA-ubiquitin (HA-Ub) was co-transfected into HEK-293 cells with FLAG-tagged arrestin3 or GRK2. After a serum starvation period of 4–6 h, the cells were treated with 10 μM DA for 1–2 min. Cellular lysates were prepared using a lysis buffer containing 150 mM NaCl, 50 mM Tris (pH 7.4), 1 mM EDTA, 1% Triton X-100, 10% (*v*/*v*) glycerol, 1 mM sodium orthovanadate, 1 mM sodium fluoride, 2 mM phenylmethylsulfonyl fluoride, 5 μg/mL leupeptin, 5 μg/mL aprotinin, and 10 mM N-ethylmaleimide. Immunoprecipitation was performed using FLAG beads, and the resulting immunoprecipitates were analyzed by SDS-PAGE, followed by blotting with antibodies against HA and FLAG, each at a dilution of 1:1000.

### 4.9. Subcellular Fractionation

Cell lysates were separated into cytoplasmic and nuclear fractions following established protocols [51,52]. Briefly, cells were incubated with buffer-1 (10 mM HEPES/KOH pH 7.8, 1.5 mM MgCl_2_, 10 mM KCl, 0.5 mM dithiothreitol, 0.2 mM phenylmethylsulfonyl fluoride, and 1 mM Na_3_VO_4_) for 20 min, then centrifuged at 2000× *g* for 5 min. The supernatant was further centrifuged at 15,000× *g* for 10 min, and this supernatant was retained as the cytoplasmic fraction.

The pellet from the initial centrifugation was washed with buffer-1 for 15 min, then centrifuged again at 15,000× *g* for 10 min. The resulting pellet was resuspended in buffer-2 (20 mM HEPES/KOH pH 7.8, 1.5 mM MgCl_2_, 420 mM NaCl, 0.2 mM EDTA, 25% glycerol, 0.5 mM dithiothreitol, 0.2 mM phenylmethylsulfonyl fluoride, and 1 mM Na_3_VO_4_) and incubated for 20 min before being centrifuged at 24,000× *g* for 10 min. The supernatant from this step was collected as the nuclear extract. Actin/GAPDH and lamin B1 were used as markers for the cytosolic and nuclear fractions, respectively.

### 4.10. Statistics

Data are presented as mean ± standard deviation (SD). To account for variability, some immunoblotting results are expressed as fold changes relative to control means. Gel density increases were determined by comparing each value to that of the mock group (typically, the first lane of the gel). Group comparisons were made using Student’s *t* test, while a one-way ANOVA with Tukey’s post hoc test was employed to compare means across multiple groups. A *p* value <0.05 was considered statistically significant.

## 5. Conclusions

We have unveiled the molecular mechanism underlying GRK2 ubiquitination and its functional implications. Our findings reveal that the ubiquitination of GRK2 induced by agonists in the cytosol requires the nuclear export of Mdm2, along with arrestin3 ubiquitination within the nucleus. Consequently, the ubiquitination processes of arrestin3 and GRK2 are intricately interconnected, with the cellular pathways governing arrestin3 ubiquitination also playing a vital role in GRK2 ubiquitination. Through this study, we have provided new insights into previously reported but challenging-to-understand molecular mechanisms.

## Figures and Tables

**Figure 1 ijms-25-09644-f001:**
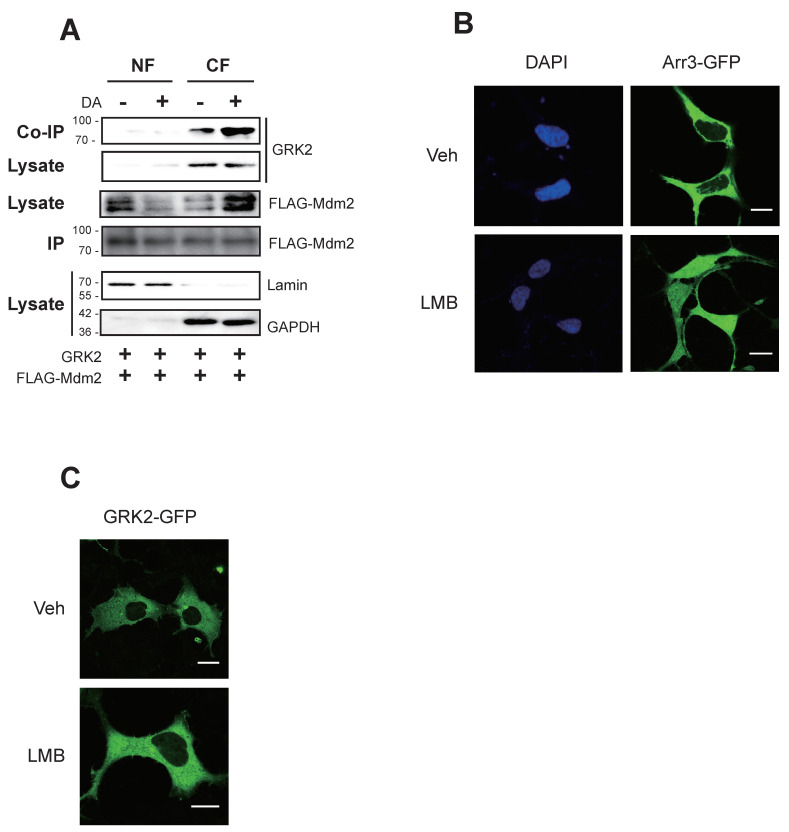
Mdm2-mediated GRK2 ubiquitination occurs in the cytoplasm. (**A**) HEK-293 cells were transfected with the D_2_R plasmid in pCDNA3.1, along with plasmids encoding GRK2 and FLAG-Mdm2. Receptor expression levels were assessed using the radioligand binding method and were maintained between 1.7 and 2.0 pmol/mg of protein. Cells were treated with 10 μM dopamine (DA) for 2 min. Subcellular fractionation was performed to separate the nuclear fraction (NF) and cytoplasmic fraction (CF). Each fraction was then immunoprecipitated using FLAG beads. The amount of immunoprecipitated FLAG-Mdm2 (IP) was adjusted to be similar in all experimental groups, then analyzed through SDS-PAGE. IP was immunoblotted with antibodies against FLAG (Mdm2). Co-IP was immunoblotted with antibodies against GRK2. Lysates were immunoblotted with antibodies against FLAG (Mdm2), GRK2, lamin, and GAPDH. When comparing the ratio of band intensities in the blot, the DA(−)/CF group (1.0 ± 0.0) showed a significant difference relative to the DA(+)/CF group (2.4 ± 0.65) (*p* < 0.01, n = 3). (**B**) HEK-293 cells expressing D_2_R (1.9–2.1 pmol/mg protein) were transfected with arrestin3-GFP. Cells were treated with 20 nM LMB for 3 h. The Pearson correlation coefficient (γ value), which represents the degree of colocalization between DAPI and arrestin3, changed from 0.27 ± 0.17 to 0.58 ± 0.25 (*p* < 0.01, n = 6) upon LMB treatment. The scale bars represent 10 μm. (**C**) HEK-293 cells expressing D_2_R (1.9–2.1 pmol/mg protein) were transfected with GRK2-GFP. Cells were treated with 20 nM LMB for 7 h. This image represents a typical example of six cells exhibiting comparable patterns. The γ value did not change significantly with LMB treatment. The scale bars represent 10 μm.

**Figure 2 ijms-25-09644-f002:**
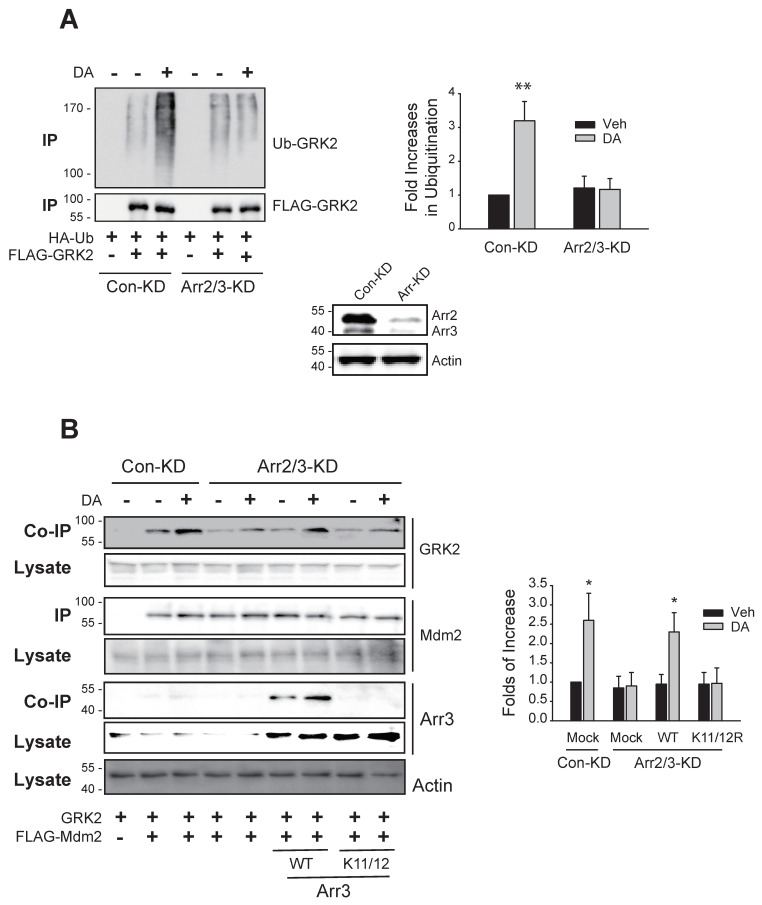
The ubiquitination of arrestin3 is needed for the ubiquitination of GRK2. (**A**) Con-KD and arrestin2/3-KD HEK-293 cells were transfected with D_2_R (1.7–2.1 pmol/mg protein), HA-Ub, and FLAG-tagged GRK2. Cells were treated with 10 µM DA for 2 min, followed by a ubiquitination assay as described in the Methods section. Immunoprecipitates were analyzed by immunoblotting with HA and FLAG antibodies. To assess knockdown efficiency, cell lysates from Con-KD and arrestin2/3-KD cells were immunoblotted with antibodies against arrestin and actin. The knockdown efficiency of arrestin2 and arrestin3 was about 90%. ** *p* < 0.01 compared to other groups (n = 3). (**B**) Con-KD and arrestin2/3-KD cells were transfected with GRK2 and FLAG-Mdm2, along with WT- or K11/12R-arrestin3. Cells were treated with 10 μM DA for 2 min. Cell lysates were immunoprecipitated with agarose beads coated with antibodies against the FLAG epitope. The amount of immunoprecipitated FLAG-Mdm2 (IP) was adjusted to be similar in all experimental groups, then analyzed through SDS-PAGE. IP was immunoblotted with antibodies against FLAG (Mdm2). Co-IPs were immunoblotted with antibodies against GRK2 and arrestin. Lysates were immunoblotted with antibodies against GRK2, FLAG (Mdm2), arrestin3, and actin. * *p* < compared to the corresponding Veh group (n = 3).

**Figure 3 ijms-25-09644-f003:**
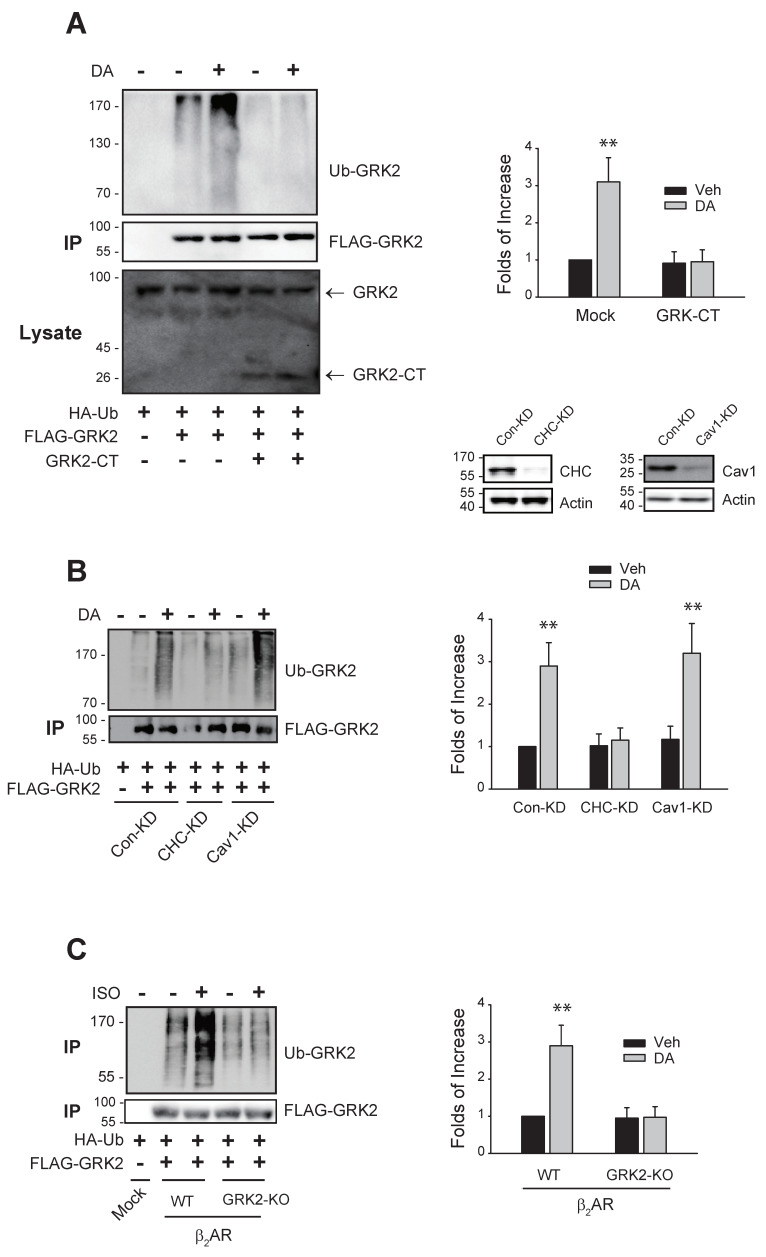
Gβγ, clathrin, and receptor phosphorylation are involved in the Mdm2-mediated ubiquitination of GRK2. Cells were treated with 10 μM DA for 2 min. Cell lysates were immunoprecipitated with FLAG beads. IPs were immunoblotted with antibodies against HA and FLAG. (**A**) HEK-293 cells expressing D_2_R (1.9–2.1 pmol/mg protein) were transfected with HA-Ub and FLAG-GRK2, along with either a mock vector or GRK2-CT. Cell lysates were immunoblotted with antibodies against GRK2. ** *p* < 0.01 compared to other groups (n = 3). GRK2-CT represents the GRK2 carboxyl terminus. (**B**) Con-KD, CHC-KD, and Cav1-KD cells were transfected with D_2_R (1.7–2.2 pmol/mg protein), HA-Ub, and FLAG-GRK2. ** *p* < 0.01 compared to each Veh group and CHC-KD cells (n = 3). To determine knockdown efficiency, the cell lysates from Con-KD, CHC-KD, and Cav1-KD cells were immunoblotted with antibodies against actin and CHC or Cav1. The knockdown efficiency of CHC and Cav1 was about 90% and 87%, respectively. CHC, clathrin heavy chain; Cav1, caveolin1. (**C**) HEK-293 cells were transfected with HA-Ub and FLAG-GRK2, together with WT-β_2_AR or GRK2-KO-β_2_AR. ** *p* < 0.01 compared to other groups (n = 3). Receptor expression levels were between 1.7 and 2.1 pmol/mg protein.

**Figure 4 ijms-25-09644-f004:**
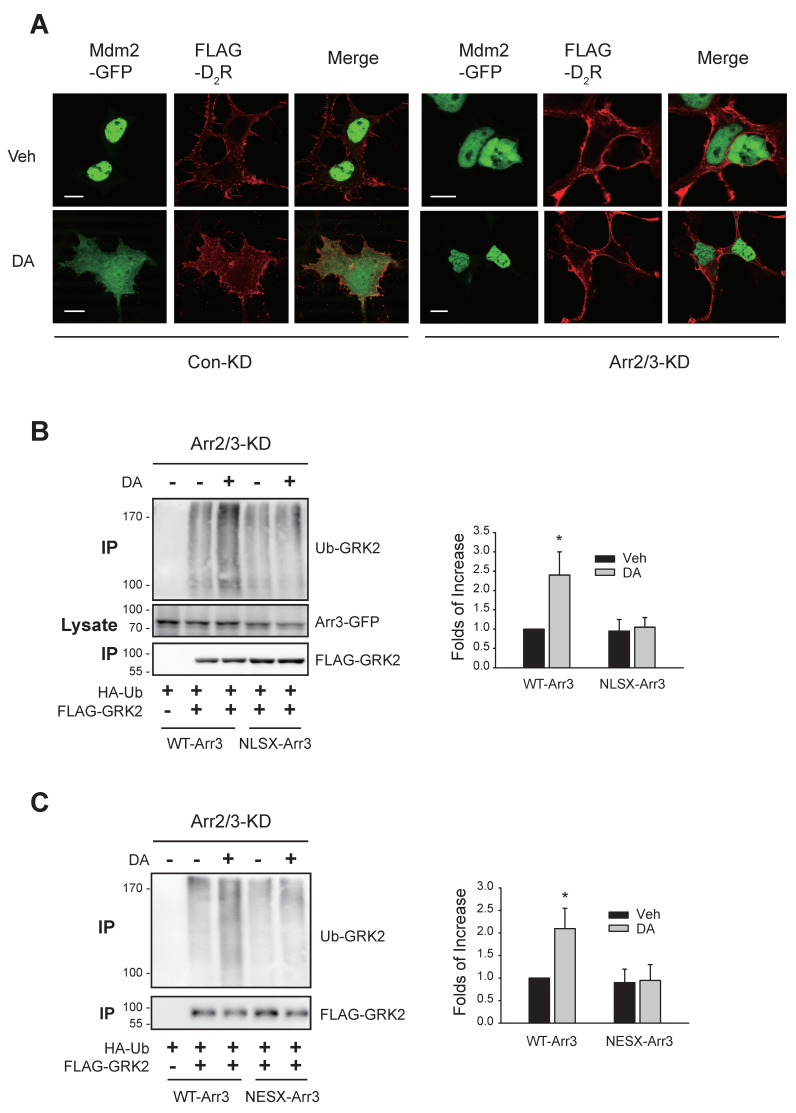
The nuclear shuttling of arrestin3 is needed to export Mdm2 and GRK2 ubiquitination. (**A**) Con-KD and arrestin2/3-KD HEK-293 cells were transfected with Mdm2-GFP and FLAG-tagged D_2_R. Cells were treated with 10 μM DA for 2 min and labeled with antibodies against FLAG. The scale bars represent 10 μm. In the Con-KD cells, the ratio of the space where Mdm2 is observed and the space surrounded by FLAG-D_2_R increased from 0.19 ± 0.03 to 0.87 ± 0.11 (*p* < 0.001, n = 7). In arrestin2/3-KD cells, the ratios were 0.30 ± 0.07 and 0.25 ± 0.07 (n = 7) for the Veh and DA groups, respectively. (**B**) Arrestin2/3-KD HEK-293 cells expressing D_2_R were transfected with HA-Ub and FLAG-GRK2, along with GFP-tagged WT-arrestin3 or NLSX-arrestin3. Cells were treated with 10 μM DA for 2 min. * *p* < 0.05 compared to the Veh/WT group (n = 3). (**C**) Arrestin2/3-KD HEK-293 cells expressing D_2_R were transfected with HA-Ub and FLAG-GRK2, along with WT-arrestin3 or NESX-arrestin3. Cells were treated with 10 μM DA for 2 min. * *p* < 0.05 compared to the Veh/WT group (n = 3).

**Figure 5 ijms-25-09644-f005:**
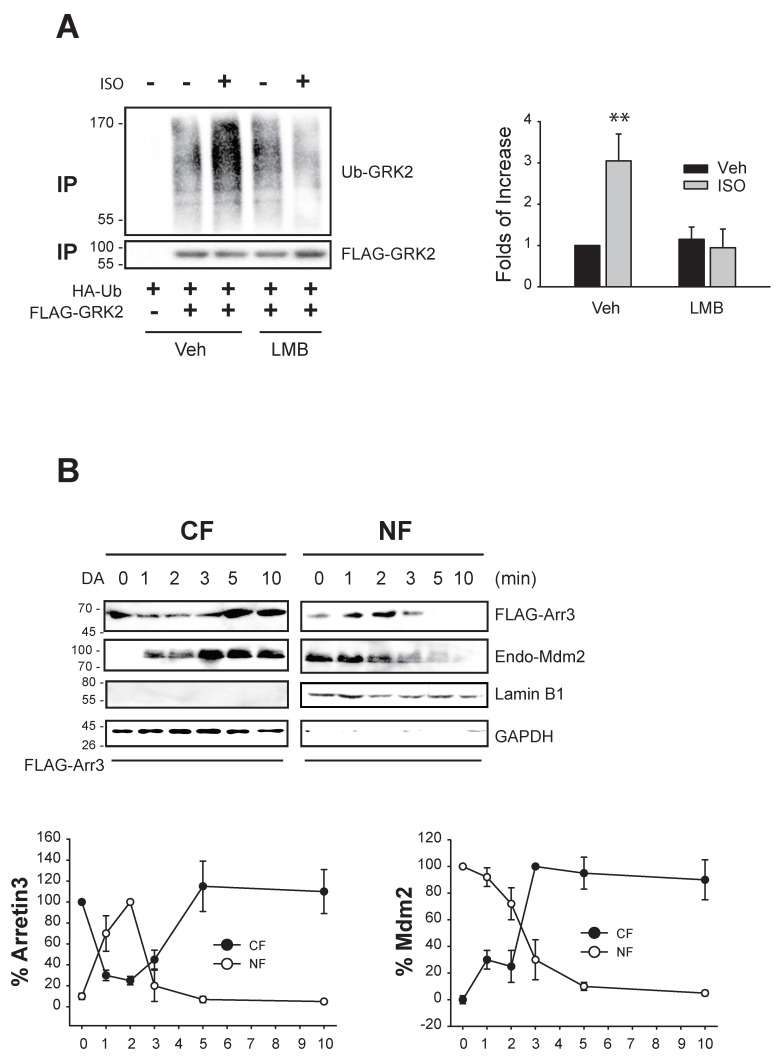
Nucleocytoplasmic traffic coupling between arrestin3 and Mdm2 mediates GRK2 ubiquitination in the cytosol. (**A**) HEK-293 cells expressing β_2_AR were transfected with HA-Ub and FLAG-GRK2. Cells were treated with vehicle or 20 nM LMB for 7 h, followed by 10 μM isoproterenol (ISO) for 2 min. ** *p* < 0.01 compared to other groups (n = 3). (**B**) HEK-293 cells expressing D_2_R were transfected with FLAG-arrestin3. Cells were treated with 10 μM DA from 0 to 10 min. Cell lysates were fractionated into cellular and nuclear fractions and immunoblotted with antibodies against FLAG, Mdm2, lamin b1, and GAPDH. The cytosolic levels of arrestin3 significantly decreased between 1 and 3 min following DA treatment (*p* < 0.001). The nuclear levels of arrestin3 significantly increased 1 and 2 min post treatment (*p* < 0.001). Both cytosolic and nuclear levels of Mdm2 exhibited significant changes 1 min after DA treatment.

**Figure 6 ijms-25-09644-f006:**
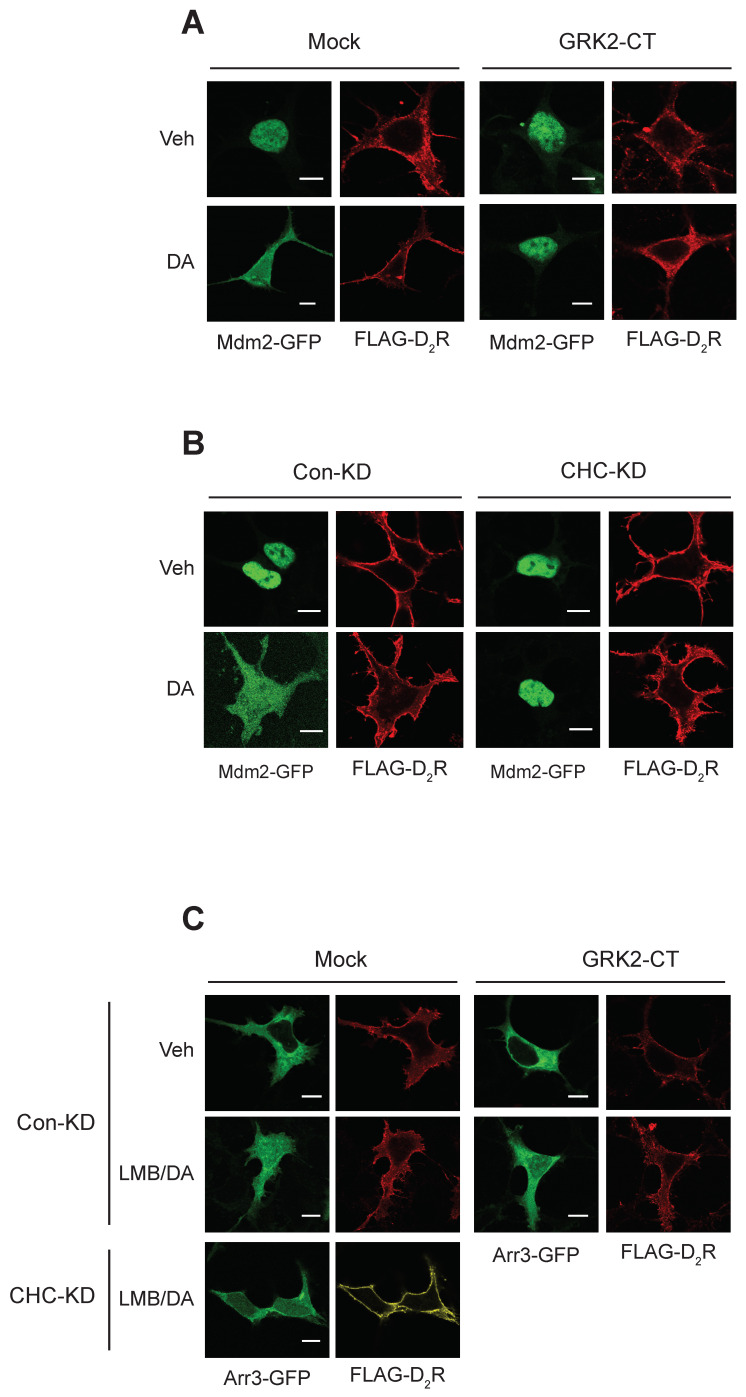
Roles of Gβγ and CHC in the nucleocytoplasmic translocation of Mdm2 and arrestin3. The results are similar for seven different cells. The scale bars represent 10 μm. (**A**) HEK-293 cells were transfected with Mdm2-GFP, along with a mock vector or GRK2-CT. Cells were treated with either vehicle or 10 μM DA for 2 min. In the mock-transfected cells, the ratio of the space where Mdm2 is observed to the space surrounded by FLAG-D_2_R increased from 0.24 ± 0.08 to 0.92 ± 0.15 (*p* < 0.001, n = 7). In the cells transfected with GRK2-CT, the ratios were 0.34 ± 0.17 and 0.37 ± 0.12 (n = 7) for the Veh and DA groups, respectively. The circumference of the cell was measured using automated software (NIS-Elements AT program; Nikon Inc., Tokyo, Japan). (**B**) Con-KD and CHC-KD HEK-293 cells were transfected with Mdm2-GFP. Cells were treated with either vehicle or 10 μM DA for 2 min. In the mock-transfected cells, the ratio of the space where Mdm2 is observed to the space surrounded by FLAG-D_2_R increased from 0.21 ± 0.07 to 0.89 ± 0.13 (*p* < 0.001, n = 7). In the CHC-KD cells, the ratios were 0.24 ± 0.18 and 0.27 ± 0.17 (n = 7) for the Veh and DA groups, respectively. (**C**) Con-KD HEK-293 cells were transfected with arrestin3-GFP, along with a mock vector or GRK2-CT (upper 8 panels); Con-KD and CHC-KD HEK-293 cells were transfected with arrestin3-GFP (left 8 panels). Cells were treated with either vehicle or 20 nM LMB for 3 h, followed by 10 μM DA for 2 min. In the Mock/Con-KD cells, the ratio of the space where Mdm2 is observed to the space surrounded by FLAG-D_2_R increased from 0.19 ± 0.08 to 0.87 ± 0.15 (*p* < 0.001, n = 8). In the GRK2-CT/Con-KD cells, the ratios were 0.21 ± 0.12 and 0.56 ± 0.42 (*p* < 0.05, n = 8) for the Veh and LMB/DA groups, respectively. In the Mock/CHC-KD cells, the ratios were 0.19 ± 0.08 and 0.27 ± 0.18 (*p* < 0.05, n= 10) for the Veh and DA/LMB groups, respectively.

**Figure 7 ijms-25-09644-f007:**
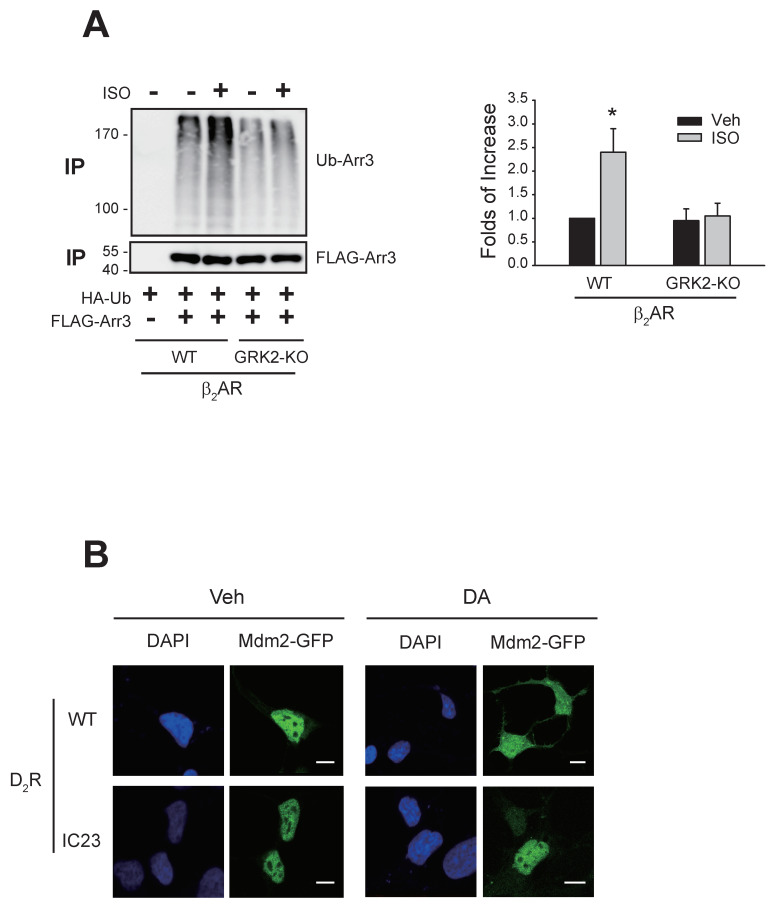
Roles of receptor phosphorylation in the ubiquitination of arrestin3 and nuclear export of Mdm2. (**A**) HEK-293 cells were transfected with HA-Ub and FLAG-arrestin3, along with WT-β_2_AR or GRK2-KO-β_2_AR. Cells were treated with 10 μM isoproterenol (ISO) for 2 min. Ubiquitination was conducted as described in the Methods section. Immunoprecipitates were immunoblotted with antibodies against HA and FLAG. * *p* < 0.05 compared to other groups (n = 3). (**B**) HEK-293 cells were transfected with Mdm2-GFP, along with WT-D_2_R or D_2_R-IC23. Cells were treated with either vehicle or 10 μM DA for 2 min. In the WT-D_2_R-expressing cells, the γ value between DAPI and Mdm2 changed from 0.95 ± 0.12 to 0.52 ± 0.15 (*p* < 0.001, n = 7) upon DA treatment. In the cells expressing IC23-D_2_R-expressing cells, the γ values were 0.93 ± 0.17 and 0.92 ± 0.15 for the vehicle- and DA-treated cells, respectively (n = 7). The scale bars represent 10 μm.

**Figure 8 ijms-25-09644-f008:**
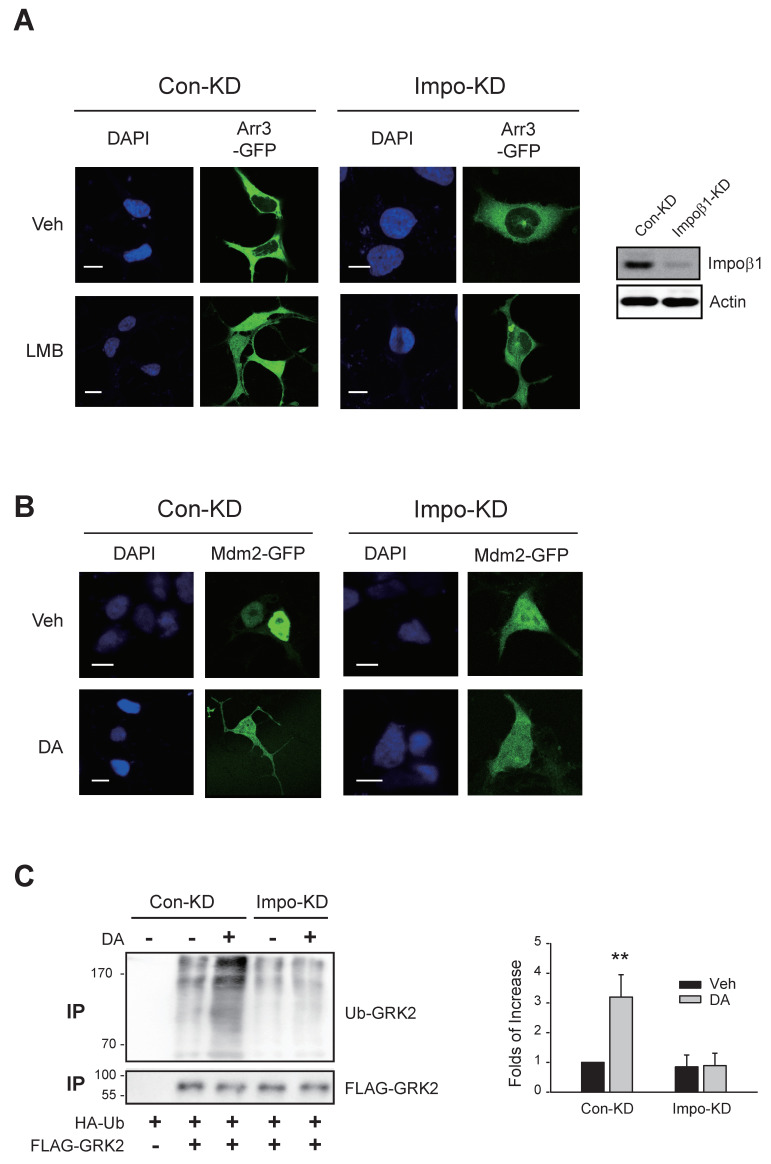
Involvement of importin complex in the nuclear entry of arrestin3 and Mdm2. The results are similar for seven different cells. The scale bars represent 10 μm. (**A**) Con-KD and importin β1-KD HEK-293 cells were transfected with arrestin3-GFP. Cells were treated with either vehicle or 20 nM LMB for 8 h. In the Con-KD cells, the γ value between DAPI and arrestin3 changed from 0.21 ± 0.08 to 0.52 ± 0.12 (*p* < 0.001, n = 7) upon LMB treatment. In the importin β1-KD cells, the γ values between DAPI and arrestin3 were 0.17 ± 0.07 and 0.24 ± 0.13 (n = 7) for the vehicle- and LMB-treated cells, respectively. The scale bars represent 10 μm. To determine the knockdown efficiency, the cell lysates from Con-KD and importin β1-KD cells were immunoblotted with antibodies against importin β1 and actin. The knockdown efficiency of importin β1 was about 93%. (**B**) Con-KD and importin β1-KD HEK-293 cells expressing D_2_R were transfected with Mdm2-GFP. Cells were treated with either vehicle or 10 μM DA. In the Con-KD cells, the γ value between DAPI and Mdm2 changed from 0.92 ± 0.18 to 0.57 ± 0.22 (*p* < 0.001, n = 7) upon LMB treatment. In the importin β1-KD cells, the γ values between DAPI and Mdm2 were 0.87 ± 0.17 and 0.91 ± 0.23 (n = 7) for the vehicle- and LMB-treated cells, respectively. The scale bars represent 10 μm. (**C**) Con-KD and importin β1-KD cells expressing D_2_R were transfected with HA-Ub and FLAG-GRK2. Cells were treated with 10 μM DA. Ubiquitination was conducted as described in the Methods section. Immunoprecipitates were immunoblotted with antibodies against HA and FLAG. ** *p* < 0.01 compared to other groups (n = 3).

**Figure 9 ijms-25-09644-f009:**
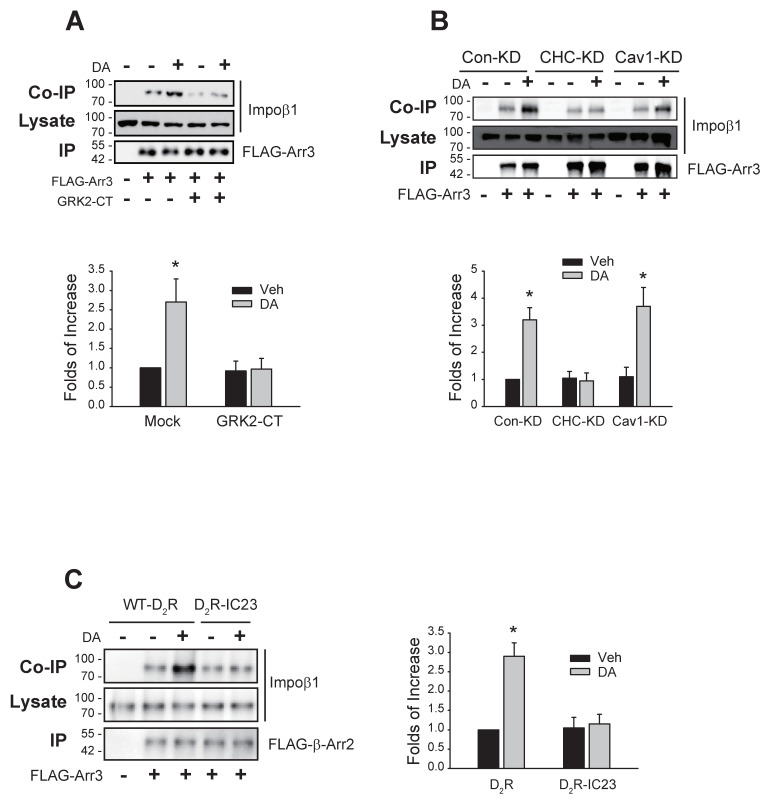
Roles of Gβγ, CHC, and receptor phosphorylation in the interaction between importin β1 and arrestin3. (**A**) HEK-293 cells expressing D_2_R were transfected with FLAG-arrestin3, along with a mock vector or GRK2-CT. Cells were treated with either vehicle or 10 μM DA for 2 min. Cell lysates were immunoprecipitated with FLAG beads. Co-IP/lysates and IPs were immunoblotted with antibodies against importin β1 and FLAG, respectively. * *p* < 0.05 compared to the Veh/Mock group (n = 3). (**B**) Con-KD, CHC-KD, and Cav1-KD HEK-293 cells transfected with D_2_R and FLAG-arrestin3. Cells were treated with either vehicle or 10 μM DA for 2 min. * *p* < 0.05 compared to the corresponding vehicle-treated group and CHC-KD cells (n = 3). (**C**) HEK-293 cells were transfected with FLAG-arrestin3, together with WT-D_2_R or D_2_R-IC23. Cells were treated with either vehicle or 10 μM DA for 2 min. * *p* < 0.05 compared to other groups (n = 3).

**Figure 10 ijms-25-09644-f010:**
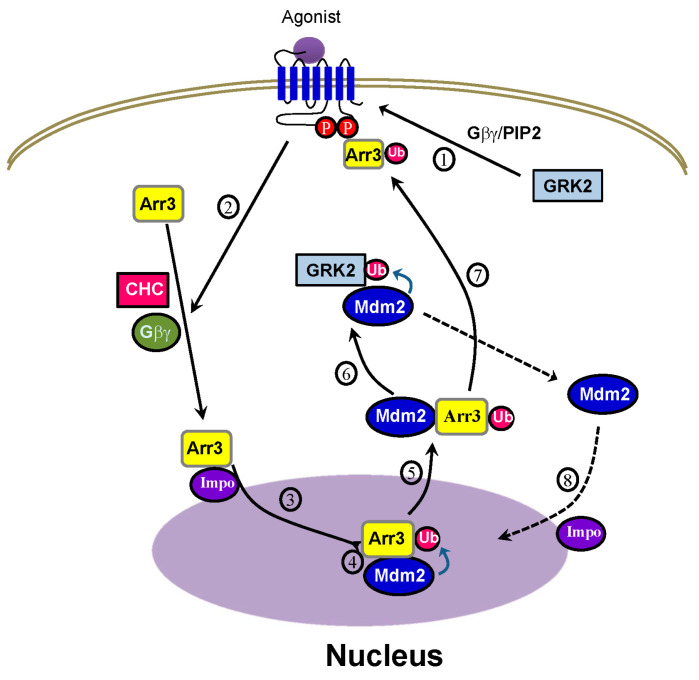
Diagram showing the interplay between arrestin3 and Mdm2 in GRK2 ubiquitination. Arrestin3 gains access to the nucleus through the importin complex (3), facilitated by the phosphorylation of receptors by GRK2, Gβγ, and clathrin (1, 2). Once in the nucleus, arrestin3 undergoes ubiquitination by Mdm2 (4) and moves to the cytoplasm, along with Mdm2 (5). In the cytosol, Mdm2 ubiquitinates GRK2 (6). Phosphorylated receptors increase their affinity for arrestins, leading to the recruitment of ubiquitinated arrestins to the activated receptors (7). Subsequently, Mdm2 returns to the nucleus through importin (8).

## Data Availability

The data that support the findings of this study are available upon request from the corresponding author.

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
