# Peer review of "The Ubiquitination of Arrestin3 within the Nucleus Triggers the Nuclear Export of Mdm2, Which, in Turn, Mediates the Ubiquitination of GRK2 in the Cytosol"

_ijms, 2024, doi:10.3390/ijms25179644_

Round 1
Reviewer 1 Report
Comments and Suggestions for Authors
The paper titled ‘The ubiquitination of arrestin3 within the nucleus triggers the nuclear export of Mdm2, which, in turn, mediates the ubiquitination of GRK2 in the cytosol’ presents a crucial topic with remarkable efficiency. The authors have demonstrated exceptional writing, experimental design, and quality of experiments. The manner in which they have established the link between multiple cellular players is outstanding. While I thoroughly enjoyed reviewing this paper, I have a few minor concerns.
Page 1, Line 31: There is a formatting error in the keywords section.
Page 4, Figure 1: The authors explain the role of MDM2 in GRK2 ubiquitination without showing the ubiquitination process. It would be beneficial to include ubiquitination of GRK2 in subcellular location to demonstrate in addition to the interaction between GRK2 and MDM2.
Page 4, Line 78: Please clarify if the cells were transfected with the b2AR plasmid.
Page 5, Line 85: What is the gamma (γ) value, needs to be explained somewhere in the paper.
General Comment: Ubiquitination is a crucial cellular process. It would be helpful if the authors could show the fate of GRK2 ubiquitination. Is GRK2 degraded by ubiquitination, or does de-ubiquitination occur via some USP?
Materials and Methods Section: The authors should include details of the cell transfection protocol.
Author Response
The paper titled ‘The ubiquitination of arrestin3 within the nucleus triggers the nuclear export of Mdm2, which, in turn, mediates the ubiquitination of GRK2 in the cytosol’ presents a crucial topic with remarkable efficiency. The authors have demonstrated exceptional writing, experimental design, and quality of experiments. The manner in which they have established the link between multiple cellular players is outstanding. While I thoroughly enjoyed reviewing this paper, I have a few minor concerns.
Q1-1) Page 1, Line 31: There is a formatting error in the keywords section.
Ans: We adjusted the font size of ‘clathrin’
Q1-2) Page 4, Figure 1: The authors explain the role of MDM2 in GRK2 ubiquitination without showing the ubiquitination process. It would be beneficial to include ubiquitination of GRK2 in subcellular location to demonstrate in addition to the interaction between GRK2 and MDM2.
Ans: Since GRK2 cannot enter the nucleus, it will be ubiquitinated by Mdm2 in the cytosol. Therefore, demonstrating the interaction between GRK2 and Mdm2 in the cytosol, as shown in Figure 1A, should be sufficient.
Q1-3) Page 4, Line 78: Please clarify if the cells were transfected with the b2AR plasmid.
Ans: Thank you for your careful review of the manuscript. The word b2AR was removed from the figure legend in Fig1A.
Q1-4) Page 5, Line 85: What is the gamma (γ) value, needs to be explained somewhere in the paper. Pearson correlation coefficient can be used to quantify the degree of colocalization between two fluorescently labeled proteins
Ans: Thank you for your helpful comment. The Pearson correlation coefficient, denoted as g, is used to quantify the degree of colocalization between two fluorescently labeled proteins. An explanation of the g value is provided in the experimental Method section (page 32). However, since Figure 1 appears earlier than the Methods section, the value is also explained in the Figure 1B legend (page 5) in the revised manuscript.
Q1-5) General Comment: Ubiquitination is a crucial cellular process. It would be helpful if the authors could show the fate of GRK2 ubiquitination. Is GRK2 degraded by ubiquitination, or does de-ubiquitination occur via some USP?
Ans: The fate of GRK2 ubiquitination has been reported to be proteasomal degradation (Penela et al., 2001). This content is included in the manuscript. In addition to this typical role in ubiquitination, we also observed that the ubiquitination of GRK2 plays an important role in the functional regulation of GPCRs. We plan to present these results in the next publication.
Q1-6) Materials and Methods Section: The authors should include details of the cell transfection protocol.
Ans: Following the reviewer's comment, the transfection process has been described in greater detail in the experimental methods section of the revised manuscript.
Reviewer 2 Report
Comments and Suggestions for Authors
Dear Authors,
I appreciate the effort and time you have dedicated to this research. However, upon reviewing the manuscript, I find that the text in several sections is not well-written and is somewhat confusing. This hinders the clarity and overall impact of your work.
Here are some specific points that need attention:
The context and literature review are insufficient, making it challenging to understand the background and significance of your study. Please expand this section to provide a more comprehensive overview.
The text accompanying the figures is unclear and inconsistent with the figure labels. For instance, it is not immediately evident that Fig 1A refers to subcellular fractionation into nuclear and cytoplasmic fractions. Additionally, some abbreviations such as "DA" are not defined. Ensure that all figures and legends are clearly labeled and consistent.
The expression levels of GRK2 and Mdm2 should be shown in total lysates to demonstrate equal expression across all conditions. Loading controls are missing in Western Blots, and protein expression levels should be shown in both IP and total lysates. The results of relative protein amounts (WB and microscopy) should be included in the main text and plotted in graphs.
The description of certain experimental results and their implications is unclear. For example, the “arrestin3 knockdown effects” and the role of the K11/12R-arrestin3 mutant need to be more clearly defined. Additionally, ensure that all experimental terms and abbreviations are clearly explained.
The Materials and Methods section needs to be clearer and more detailed. Include sequences for plasmid DNA constructs and shRNAs, specify antibodies used for immunofluorescence, and provide more information on image acquisition.
The authors described in the main text the role of arrestin3 but on the figures is indicated Arr2/3 knockdown, please be more clear and specific.
Use additional markers to exclude contamination in subcellular fractions; β-actin and Lamin B1 are insufficient, the authors cannot exclude the potential contamination with different organelles.
Levels of protein ubiquitination should be assessed with endogenous levels of ubiquitin, overexpression of HA-Ub is not necessary.
Some specific examples are described below:
Figure 1:
Text should be clearer and use the same denominations of the figures, for example is not clear that Fig1A corresponds to subcellular fractionation into nuclear fraction (NF) and cytoplasmic fraction (CF), and there is indication of what DA means.
Expression of GRK2 and Mdm2 in the total lysate should be shown, to demonstrate equal expression in all conditions.
Ensure that the labelling for the shown co-IP and IP, as well as the labelling for Lamin and β-actin, are not from the same membrane.
Immunocytochemistry (line 72) implies the use of antibodies to recognize the protein, but Fig.1B and 1C depict GFP expression.
The relative amounts of proteins (WB and microscopy) described in the figure legend should be on the main text and plotted in a graph.
Figure 2:
Knockdown efficiency should be showed by WB of arrestin 3 in total lysates in the same samples used for IP.
Levels of FLAG-GRK2 should be shown also in the IP samples and not only on total lysates.
Rewrite the text for clarity: specify the “arrestin3 knockdown effects” and explain the significance of the K11/12R-arrestin3 mutant.
Clarify the meaning and function/effect of "DA."
Is not clear what was quantified and plotted in the graph of Fig2B.
Figure 3:
Knockdown efficiency should be showed.
Levels of protein expression under study should be shown not only in IP but also in the total lysates.
There is no mention of Cav1 KD in the main text.
Figure 4:
Localization of arrestin3 mutants should be shown, as well as its impact on Mdm2 and GRK2 localization/distribution.
Figure 8:
Knockdown of importin should be shown in the total lysates (fig8C).
Additionally, the novelty of the study is limited, it lacks some crucial results and controls, has serious methodological flaws, and the statements and conclusions made by the authors are not fully supported by the data presented.
Given these points, I recommend a thorough revision of the manuscript to improve clarity and coherence. This will greatly enhance the readability and impact of your research findings.
After careful consideration, I regret to inform you that I do not recommend the revision or publication of this manuscript in its current form. Therefore, my recommendation is for rejection.
Comments on the Quality of English LanguageI find that the text in several sections is not well-written and is somewhat confusing. This hinders the clarity and overall impact of your work.
Author Response
Dear Authors,
I appreciate the effort and time you have dedicated to this research. However, upon reviewing the manuscript, I find that the text in several sections is not well-written and is somewhat confusing. This hinders the clarity and overall impact of your work.
Here are some specific points that need attention:
Q2-1) The context and literature review are insufficient, making it challenging to understand the background and significance of your study. Please expand this section to provide a more comprehensive overview.
Ans: Other reviewers provided similar feedback regarding the logical clarity of the text in several sections. Consequently, the manuscript has been revised according to these comments to enhance its readability and ensure it is as easy as possible for readers to understand.
Q2-2) The text accompanying the figures is unclear and inconsistent with the figure labels. For instance, it is not immediately evident that Fig 1A refers to subcellular fractionation into nuclear and cytoplasmic fractions. Additionally, some abbreviations such as "DA" are not defined. Ensure that all figures and legends are clearly labeled and consistent.
Ans: Other referees provided similar feedback, so the figure legends and text were modified accordingly.
Q2-3) The expression levels of GRK2 and Mdm2 should be shown in total lysates to demonstrate equal expression across all conditions. Loading controls are missing in Western Blots, and protein expression levels should be shown in both IP and total lysates. The results of relative protein amounts (WB and microscopy) should be included in the main text and plotted in graphs.
Ans: First, we agree with the referee's comment that the lysate should be shown. However, in the case of the ubiquitination assay, since ubiquitination is quantified using IPed proteins, we believe that showing both lysate and IP is somewhat redundant. Second, we also agree with the comments about loading control. Generally, the interaction between proteins is considered significant when the amount of IP is more than 1% of the total lysate. Such strict control is necessary when the interaction between the two proteins is not yet clear. On the other hand, since the Co-IP in this study involves proteins with a well-known interaction, the absence of a loading control does not appear to significantly impact data interpretation.
Q2-4) The description of certain experimental results and their implications is unclear. For example, the “arrestin3 knockdown effects” and the role of the K11/12R-arrestin3 mutant need to be more clearly defined. Additionally, ensure that all experimental terms and abbreviations are clearly explained.
Ans: Other referees provided similar feedback, so the figure legends and text were modified accordingly.
Q2-5) The Materials and Methods section needs to be clearer and more detailed. Include sequences for plasmid DNA constructs and shRNAs, specify antibodies used for immunofluorescence, and provide more information on image acquisition.
Ans: Plasmid DNA sequences can be easily accessed from gene banks, and there is insufficient space in the manuscript to present these sequences. The shRNA used to create knockdown cell lines was purchased from Promega, but due to company policy, the sequence was not provided. Additionally, the RRID for antibodies has been included following the referee's comments.
Q2-6) The authors described in the main text the role of arrestin3 but on the figures is indicated Arr2/3 knockdown, please be more clear and specific.
Ans: A common issue with experiments involving only the knockdown of arrestin 3 is that arrestin 2 might compensate for its role, making the results less clear. In contrast, knocking down both arrestin 2 and arrestin 3 and then reintroducing them separately would provide a clearer insight into the distinct roles of each subtype.
Q2-7) Use additional markers to exclude contamination in subcellular fractions; β-actin and Lamin B1 are insufficient, the authors cannot exclude the potential contamination with different organelles.
Ans: Thank you for the helpful suggestion. However, these markers are commonly used to label the cytosol and nucleus. We also used actin as a cytosolic marker but did not observe any difference from GAPDH.
Q2-8) Levels of protein ubiquitination should be assessed with endogenous levels of ubiquitin, overexpression of HA-Ub is not necessary.
Ans: Thank you for the good suggestion. However, the protocol used in this study is the most widely used method for protein ubiquitination assay.
Some specific examples are described below:
Q2-9) Figure 1:
- Text should be clearer and use the same denominations of the figures, for example is not clear that Fig1A corresponds to subcellular fractionation into nuclear fraction (NF) and cytoplasmic fraction (CF), and there is indication of what DA means.
Ans: The sentences were revised based on the referee's suggestions to improve clarity (page 3). For DA, the full name "dopamine" was provided in the figure legend to aid understanding.
- Expression of GRK2 and Mdm2 in the total lysate should be shown, to demonstrate equal expression in all conditions.
Ans: (During the revision process and discussions with co-authors, we discovered a labeling mistake. The part labeled ‘Lysate’ should have been ‘IP’. The figure has been corrected accordingly.) The principle of co-immunoprecipitation is to determine the extent to which the IPed Mdm2 interacts with GRK2. In this context, the crucial factor is the amount of IP, so we believe the results can be interpreted correctly even without showing the lysate. Furthermore, for GRK2, we performed co-immunoprecipitation of the endogenous protein. Therefore, we do not anticipate any differences in GRK2 expression between the groups.
- Ensure that the labelling for the shown co-IP and IP, as well as the labelling for Lamin and β-actin, are not from the same membrane.
Ans: They are obtained all from the same sample and the same membrane.
- Immunocytochemistry (line 72) implies the use of antibodies to recognize the protein, but Fig.1B and 1C depict GFP expression.
Ans: We changed ‘immunocytochemistry’ to ‘fluorescence microscopy’.
- The relative amounts of proteins (WB and microscopy) described in the figure legend should be on the main text and plotted in a graph.
Ans: Of course, that may also be a good way to explain the experimental results. However, we believe it would be more effective to include these numbers in the figure legend, allowing readers to refer to them while viewing the figures.
Q2-10) Figure 2:
- Knockdown efficiency should be showed by WB of arrestin 3 in total lysates in the same samples used for IP.
Ans: Yes, we used the same stable knockdown cell line.
- Levels of FLAG-GRK2 should be shown also in the IP samples and not only on total lysates.
Ans: (During the revision process and discussions with co-authors, we discovered a labeling mistake. The part labeled ‘Lysate’ should have been ‘IP’. The figure has been corrected accordingly.) The principle of ubiquitination is determining the extent to which the IPed GRK2 is ubiquitinated. In this context, the crucial factor is the amount of IP, so we believe the results can be interpreted correctly without showing the lysate.
- Rewrite the text for clarity: specify the “arrestin3 knockdown effects” and explain the significance of the K11/12R-arrestin3 mutant.
Ans: It was changed as follows. ‘The ubiquitination of GRK2, which was abolished by arrestin knockdown, was restored by co-expression of WT-arrestin3, but not by K11/12R-arrestin3’. In addition, we added an explanation for K11/12R-arrestin3.
- Clarify the meaning and function/effect of "DA."
Ans: It is difficult for us to figure out exactly what the referee wants.
- Is not clear what was quantified and plotted in the graph of Fig2B.
Ans: Fig. 2B illustrates the interaction between Mdm2 and GRK2 following treatment with either vehicle or DA across four experimental groups. The bar graph depicts the Co-IP ratio between Mdm2 and GRK2 induced by vehicle and DA treatments.
Q2-11) Figure 3:
- Knockdown efficiency should be showed.
Ans: The knockdown efficiency is shown in the figure legend of Fig. 3B.
- Levels of protein expression under study should be shown not only in IP but also in the total lysates.
Ans: (During the revision process and discussions with co-authors, we discovered a labeling mistake. The part labeled ‘Lysate’ should have been ‘IP’. The figure has been corrected accordingly.) The principle of ubiquitination is to determine the extent to which the IPed GRK2 is ubiquitinated. In this context, the crucial factor is the amount of IP, so I believe the results can be interpreted correctly even without showing the lysate.
- There is no mention of Cav1 KD in the main text.
Ans: We took care of it (page 7).
Q2-12) Figure 4:
Localization of arrestin3 mutants should be shown, as well as its impact on Mdm2 and GRK2 localization/distribution.
Ans: Most of the information above comment is shown in our previous publication (Zhang et al., 2020). Your comments are very good but at the same time so extensive that explaining everything systematically would require writing an entirely new paper.
Q2-13) Figure 8:
- Knockdown of importin should be shown in the total lysates (fig8C).
Ans: As shown in Fig. 8A, the cellular level of importin b1 was determined from the total cell lysates. For clarity, we included the following sentence. “Cell lysates from Con-KD and importin b1-KD cells were immunoblotted with antibodies against importin b1 and actin.”
Reviewer 3 Report
Comments and Suggestions for Authors
Kundu and colleagues present a manuscript where they decipher molecular mechanisms of GRK2 ubiquitinylation. They show that upon stimulation arrestin3 first translocates to the nucleus, gets ubiquitinylated there by Mdm2, then leaves the nucleus together with Mgm2. Both ubiquitinylate GRK2, which in turn, together with arrestin3, binds to the activated GPCR on the plasma membrane. The topic is very interesting, the study is overall well performed and the data quite convincing. However, the manuscript appears a bit sloppy and needs several points to be addressed before publication.
1. The sentence “agonist-induced….” appears twice in the abstract
2. line 48ff, “A previous study….” Ref. 8, is that from your lab? If yes you could state that.
3. l. 55ff “both GRK” and arrestin are ubi….” You state that already in l. 37ff.
4. legend Fig. 1: “was conducted into” sounds a bit strange, better: subcellular fractionation was performed to separate the nuclear fraction (NF) and cytoplasmic fraction (CF). Each fraction was then immunoprecipitated using FLAG beads.
5. legend Fig. 1 Co-IP and IP: sounds as if there were two IPs performed, but as I understand one IP was done, with FLAG-beads, and blotted for FLAG and for GRK2. So just say Co-IP.
6. legend Fig. 1, DA. What is DA? Please indicate here, it only becomes clear much later that this in dopamine.
7. legend Fig. 1: DA(+)/CF group (1.0 ± 0.0) was significantly different from 82 DA(-)/CF group (2.4 ± 0.65) . What does that mean, is it the ratio of band intensities in the blot? Please explain
8. legend Fig. 1B: what is the gamma-value? Please define
9. l. 97 K11/12R-arrestin3, what is that? Hasn´t been introduced yet
10. Fig.2: lanes 2/3 and 5/6 are labeled identically with pluses. I guess lanes 2 and 5 are without DA and should be labeled -. In the legend of A it is not indicated that an IP was performed, nor which one. FLAG, I guess. In the graph on the right, what does the y-axis refer to? Fold increase of what?
11. Fig. 2A legend (also refers to other figures). Why do you indicate the concentration of D2R? First, the phrasing “transfected with D2R (1.7-2.1 pmol/mg protein) suggests you transfected protein, but I guess you transfected a plasmid coding for D2R. Second, what does this concentration tell, is it an indicator of overexpression, or similar expression to the endogenous? If not explained, these values are meaningless.
12. Fig. 3 and accompanying text. Fig. is labelled with CHC-KD and Cav1-KD, but one has to guess that CHC refers to clathrin (precisely clathrin heavy chain). Caveolin is not even mentioned. Also, GRK2-CT is used in the Fig., but it remains mysterious what that could be? That veh means vehicle could be mentioned at one point. Finally, in legend to 3C the Co-IP is not mentioned.
13. In the context of Fig. 4 arrestin3 WT and mutants are re-expressed in an arrestin knock-down background. Are the constructs resistant to the co-transfected siRNA (whose sequence(s) by the way should be indicated in the methods)? Or does the siRNA target regions outside the coding region?
14. Fig. 5B. These results could be nicely backed-up by immunofluorescence or even better live-cell microscopy. Legend Fig. 5A: what´s ISO?
15. Fig. 6A. For the ratio only 7 cells were used? From how many experiments, only 1? That sounds not very convincing.
16. Fig. 6. Replace “horizontal” by “scale”
17. l. 254: Isn´t that sentence a circular reasoning? If receptor phosphorylation is necessary for ubi of arrestin and so on, and if all that is required for receptor phosphorylation then nothing ever happens, because one is the essential precondition for the other. There must be a way out of that. Please reconsider.
18. l. 278. Why arrestin3 needs a partner with an NLS if it has one on its own?
19. Discussion: the transition between the paragraph lines 355-359 to the next reads a bit awkward.
20. Fig. 10. Did you really show that arrestin3 is ubiquitinylated in the nucleus? For this you need to use the export-defective arrestin-mutant and analyze its ubiquitinylation, which I didn´t see in the manuscript.
21. Discussion about clathrin. That is not yet satisfying, I don´t think the data are suggesting an additional role for clathrin beyond endocytosis. You did not discuss the role of endocytosis of the receptor that could induce down-stream processes, for example step2 in your model in Fig. 10. KD of clathrin would then simply inhibit endocytosis of the receptor and block further events. Something along that line should be discussed.
Author Response
Kundu and colleagues present a manuscript where they decipher molecular mechanisms of GRK2 ubiquitinylation. They show that upon stimulation arrestin3 first translocates to the nucleus, gets ubiquitinylated there by Mdm2, and then leaves the nucleus together with Mgm2. Both ubiquitinylate GRK2, which in turn, together with arrestin3, binds to the activated GPCR on the plasma membrane. The topic is very interesting, the study is overall well performed and the data is quite convincing. However, the manuscript appears a bit sloppy and needs several points to be addressed before publication.
Q3-1. The sentence “agonist-induced….” appears twice in the abstract
Ans: Thank you for your careful review of the manuscript. The manuscript was revised according to the referee's advice.
Q3-2. line 48ff, “A previous study….” Ref. 8, is that from your lab? If yes you could state that.
Ans: Yes, it is from my lab.
Q3-3. l. 55ff “both GRK” and arrestin are ubi….” You state that already in l. 37ff.
Ans: Thank you for your careful review of the manuscript. Because this sentence was repeated in two places, the latter part was removed.
Q3-4. legend Fig. 1: “was conducted into” sounds a bit strange, better: subcellular fractionation was performed to separate the nuclear fraction (NF) and cytoplasmic fraction (CF). Each fraction was then immunoprecipitated using FLAG beads.
Ans: The manuscript was revised as suggested by the referee.
Q3-5. legend Fig. 1 Co-IP and IP: sounds as if there were two IPs performed, but as I understand one IP was done, with FLAG-beads, and blotted for FLAG and for GRK2. So just say Co-IP.
Ans: In this context, IP refers to the FLAG-tagged protein Mdm2, while Co-IP refers to the interacting protein, GRK2, which is immunoprecipitated along with Mdm2.
Q3-6. legend Fig. 1, DA. What is DA? Please indicate here, it only becomes clear much later that this in dopamine.
Ans: In response to the referee's comment, the sentence "Cells were treated with 10 µM dopamine (DA)" has been added to the legend of Figure 1A.
Q3-7. legend Fig. 1: DA(+)/CF group (1.0 ± 0.0) was significantly different from DA(-)/CF group (2.4 ± 0.65). What does that mean, is it the ratio of band intensities in the blot? Please explain.
Ans: Thank you for your comment. Without your comment, we would have made a huge mistake. The sentence was changed to “When comparing the ratio of band intensities in the blot, the DA(-)/CF group (1.0 ± 0.0) showed a significant difference from the DA(+)/CF group (2.4 ± 0.65) (p < 0.01, n = 3).”
Q3-8. legend Fig. 1B: what is the gamma-value? Please define
Ans: Thank you for your helpful comment. The Pearson correlation coefficient, denoted as g, is used to quantify the degree of colocalization between two fluorescently labeled proteins. The explanation of the g value is provided in the experimental Method section (page 31). However, since Figure 1 appears earlier than the Methods section, the value is also explained in the Figure 1B legend (page 5) in the revised manuscript.
Q3-9. l. 97 K11/12R-arrestin3, what is that? Hasn´t been introduced yet
Ans: The following sentence was added to the revised manuscript. “K11/12R-arrestin3 is unable to bind to Mdm2 due to a mutation in the Mdm2 binding site (Shenoy and Lefkowitz, 2005).”
Q3-10. Fig.2: lanes 2/3 and 5/6 are labeled identically with pluses. I guess lanes 2 and 5 are without DA and should be labeled -. In the legend of A it is not indicated that an IP was performed, nor which one. FLAG, I guess. In the graph on the right, what does the y-axis refer to? Fold increase of what?
Ans: Thank you for your comment. Without your comment, we would have made a huge mistake.
- i) Yes. The label was changed to (-).
- The protocol for ubiquitination is described in detail in the method section. Thus, the sentence was changed to ‘Cells were treated with 10 µM DA for 2 min, and the ubiquitination assay was performed as outlined in the Methods section.’
- The labeling for the y-axis was changed to ‘Fold Increases in Ubiquitination’.
Q3-11. Fig. 2A legend (also refers to other figures). Why do you indicate the concentration of D2R? First, the phrasing “transfected with D2R (1.7-2.1 pmol/mg protein) suggests you transfected protein, but I guess you transfected a plasmid coding for D2R. Second, what does this concentration tell, is it an indicator of overexpression, or similar expression to the endogenous? If not explained, these values are meaningless.
Ans: Cellular responses are largely dictated by receptor expression levels. Thus, when studying downstream events associated with receptors, it is essential to maintain comparable receptor expression levels across different experimental groups.
Q3-12. Fig. 3 and accompanying text. Fig. is labeled with CHC-KD and Cav1-KD, but one has to guess that CHC refers to clathrin (precisely clathrin heavy chain). Caveolin is not even mentioned. Also, GRK2-CT is used in the Fig., but it remains mysterious what that could be? That veh means vehicle could be mentioned at one point. Finally, in legend to 3C the Co-IP is not mentioned.
Ans: i) CHC and Cav1 are explained on page 7, where they are first introduced in the text. CHC (clathrin heavy chain) and Cav1 (caveolin1) are also shown in the Fig.3B legend.
- ii) GRK2-CT (GPCR kinase carboxyl-terminus) is explained on page 14, where it is first introduced in the text. This is also shown in the legend of Fig.3A.
iii) During the revision process and discussions with co-authors, we discovered a labeling mistake. The part labeled ‘Lysate’ should have been ‘IP’. The figure has been corrected accordingly. The sentence "Cells were treated with 10 μM DA for 2 minutes. Cell lysates were immunoprecipitated with FLAG beads. Co-IP/IP were immunoblotted with antibodies against HA and FLAG." applies to all figures, so it was moved to the beginning of the figure legend.
Q3-13. In the context of Fig. 4 arrestin3 WT and mutants are re-expressed in an arrestin knock-down background. Are the constructs resistant to the co-transfected siRNA (whose sequence(s) by the way should be indicated in the methods)? Or does the siRNA target regions outside the coding region?
Ans: The arrestin knocked down with shRNA is human arrestin, while the introduced one is rat arrestin. Despite their sequence similarities, there is no significant issue even if the shRNA degrades the transfected arrestin. This is because the shRNA is stably transfected and expressed at low levels, whereas the transiently transfected arrestin is expressed at much higher levels. Consequently, the higher expression of transiently transfected arrestin can adequately compensate for any knocked-down arrestin.
Q3-14. Fig. 5B. These results could be nicely backed-up by immunofluorescence or even better live-cell microscopy. Legend Fig. 5A: what´s ISO?
Ans: Thank you for the great suggestion. We will try this method the next time we conduct a similar experiment. ISO represents isoproterenol. We included this in the figure legend.
Q3-15. Fig. 6A. For the ratio only 7 cells were used? From how many experiments, only 1? That sounds not very convincing.
Ans: I concur with the referee's observation. We also noted this issue and verified it through two independent experiments using standard fluorescence microscopy. Subsequently, we captured high-quality images with confocal microscopy and performed statistical analysis of the data.
Q3-16. Fig. 6. Replace “horizontal” by “scale”
Ans: We changed ‘horizontal’ to ‘scale’.
Q3-17. l. 254: Isn´t that sentence a circular reasoning? If receptor phosphorylation is necessary for ubi of arrestin and so on, and if all that is required for receptor phosphorylation then nothing ever happens, because one is the essential precondition for the other. There must be a way out of that. Please reconsider.
Ans: Thank you for your insightful comments. Many receptors, including GPCRs, undergo simultaneous signaling and regulation processes when stimulated by agonists. Strong amplification at each step of receptor signaling is crucial for the efficient operation of the signaling system. GRK2 and arrestin3 play key roles in the regulation of GPCRs. Similar to the signaling process, the regulation process likely requires amplification at each step for efficient functional performance. However, there is a lack of mechanistic understanding of how this regulatory process is amplified. In this context, we believe that the positive correlation between the interdependence of GRK2 and arrestin3 ubiquitination and receptor phosphorylation observed in this study provides an important perspective for understanding the mechanism of GPCR regulation. We have included these details in the discussion section.
Q3-18. l. 278. Why arrestin3 needs a partner with an NLS if it has one on its own?
Ans: For a protein to enter the nucleus through the nuclear pore complex, it must contain a nuclear localization signal (NLS). The protein gains entry by binding to the importin complex, which recognizes and attaches to the NLS. This process is facilitated by Ran, a small G protein that uses GTP as its energy source.
Q3-19. Discussion: the transition between the paragraph lines 355-359 to the next reads a bit awkward.
Ans: Thank you for your helpful comments on the manuscript. In accordance with the comments, two paragraphs of the discussion have been modified so that the authors can more easily understand the purpose of this study.
Q3-20. Fig. 10. Did you really show that arrestin3 is ubiquitinylated in the nucleus? For this you need to use the export-defective arrestin-mutant and analyze its ubiquitinylation, which I didn´t see in the manuscript.
Ans: Yes, we confirmed it. This issue has been thoroughly explored in a paper we previously published (Zhang et al., 2020).
Q3-21. Discussion about clathrin. That is not yet satisfying, I don´t think the data are suggesting an additional role for clathrin beyond endocytosis. You did not discuss the role of endocytosis of the receptor that could induce downstream processes, for example, step2 in your model in Fig. 10. KD of clathrin would then simply inhibit endocytosis of the receptor and block further events. Something along that line should be discussed.
Ans: Thank you for your insightful comments. The role of clathrin in receptor endocytosis is so critical that many researchers find it difficult to consider clathrin functioning beyond coating endocytic vesicles. As you mentioned, receptor endocytosis might alter the structure depicted in Fig. 10 by reducing the number of receptors on the plasma membrane. However, GRK2 ubiquitination peaks 2 minutes after agonist stimulation, while receptor endocytosis becomes apparent at least 20 minutes after agonist treatment. Therefore, receptor endocytosis is not expected to significantly impact the composition shown in Fig. 10.
Reviewer 4 Report
Comments and Suggestions for Authors
The research manuscript by Kundu et al for International Journal of Molecular Sciences is well written and interesting manuscript.
I would like to raise following comments:
1. Please provide a short introduction of GRK2 too like that of arrestin3 in line 35-36.
2. Provide reference for GRK2 does not posses NLS in page 51
3. Are Mdm2 only ubiquitin ligase for arrestin3 and GRK2?
4. If the ubiquitination of GRK2 is followed by that of arrestin3, does author see any delay in the ubiquitination of GRK2 with that of arrestin3?
5. In the discussion part, please provide a short description of the importance of arrestin3 and GRK2 ubiquitination in the context of GPCR signaling.
6. During DA treatment, figure 5B, there is dramatic increase in the level of Mdm2 and arrestin3 in the cytosolic fraction compared with nuclear fraction. Doesn’t it mean, Mdm2 translocation to cytosol and arrestin3 ubiquitination occur predominantly in cytosol in compared to nucleus? Also, protein ubiquitination and proteasomal degradation occur in the cytosol, how do author reconcile this with their finding?
Author Response
The research manuscript by Kundu et al for International Journal of Molecular Sciences is well written and interesting manuscript. I would like to raise following comments:
Q4-1. Please provide a short introduction of GRK2 too like that of arrestin3 in line 35-36.
Ans: Based on the referee's suggestion, the introduction part was modified as follows. Following agonistic stimulation of G protein-coupled receptors (GPCRs), Gα and Gβγ subunits of the associated heterotrimeric G protein dissociate (Gilman, 1987). The released Gβγ subunit, together with plasma membrane phospholipids, binds to the pleckstrin homology (PH) domain of GPCR kinase 2 (GRK2) located in the carboxyl terminus region. This binding recruits the inactive cytoplasmic GRK2 to the plasma membrane, where agonist-bound GPCRs are located (Pitcher et al., 1996; Pitcher et al., 1992; Touhara et al., 1994). GRK2 phosphorylates the receptors to provide high-affinity sites for arrestins (Benovic et al., 1987; Benovic et al., 1986).
Q4-2. Provide reference for GRK2 does not posses NLS in page 51 (line 51)
Ans: Yes, we provided a related reference.
Q4-3. Are Mdm2 only ubiquitin ligase for arrestin3 and GRK2?
Ans: There have been reports suggesting that NEDD4 and SMURF2 interact with arrestin, and c-Cbl and CHIP interact with GRK2. However, their roles as E3 ligases for arrestin and GRK2 have not been clearly demonstrated.
Q4-4. If the ubiquitination of GRK2 is followed by that of arrestin3, does author see any delay in the ubiquitination of GRK2 with that of arrestin3?
Ans: Our results indicate that the ubiquitination of arrestin3 and GRK2 is closely interconnected, mutually enhancing each other. Therefore, if we could observe their interaction on very short timescales, we might detect a time difference between the ubiquitination events of arrestin3 and GRK2. However, since these processes are already in dynamic equilibrium, capturing this time gap will be challenging.
Q4-5. In the discussion part, please provide a short description of the importance of arrestin3 and GRK2 ubiquitination in the context of GPCR signaling.
Ans: According to the referee's comment, the following content was added to the discussion.
“Many receptors, including GPCRs, undergo simultaneous signaling and regulation processes when stimulated by agonists. Strong amplification at each step of receptor signaling is crucial for the efficient operation of the signaling system. GRK2 and arrestin3 play key roles in the regulation of GPCRs. Similar to the signaling process, the regulation process likely requires amplification at each step for efficient functional performance. However, there is a lack of mechanistic understanding of how this regulatory process is amplified. In this context, we believe that the positive correlation between the interdependence of GRK2 and arrestin3 ubiquitination and receptor phosphorylation observed in this study provides an important perspective for understanding the mechanism of GPCR regulation.”
Q4-6. During DA treatment, figure 5B, there is dramatic increase in the level of Mdm2 and arrestin3 in the cytosolic fraction compared with nuclear fraction. Doesn’t it mean, Mdm2 translocation to cytosol and arrestin3 ubiquitination occur predominantly in cytosol in compared to nucleus? Also, protein ubiquitination and proteasomal degradation occur in the cytosol, how do author reconcile this with their finding?
Ans: It was reported through various mechanistic studies that arrestin3 is ubiquitinated by Mdm2 in the nucleus (Zhang et al., 2020). However, ubiquitination that occurs within the nucleus refers to the agonist-induced increase in ubiquitination, not basal ubiquitination. Ubiquitinated arrestin3 within the nucleus will move to the cytosol probably through a concentration-dependent manner, and then translocate to the plasma membrane or undergo proteasomal degradation.
In addition, if Mdm2 moves to the cytosol for some reason, it will interact with arrestin3 and basal ubiquitination will occur in the cytosol (Min et al., 2017; Min et al., 2023; Zheng et al., 2020a; Zheng et al., 2020b).
Benovic, J.L., H. Kuhn, I. Weyand, J. Codina, M.G. Caron, and R.J. Lefkowitz. 1987. Functional desensitization of the isolated beta-adrenergic receptor by the beta-adrenergic receptor kinase: potential role of an analog of the retinal protein arrestin (48-kDa protein). Proc Natl Acad Sci U S A. 84:8879-8882.
Benovic, J.L., R.H. Strasser, M.G. Caron, and R.J. Lefkowitz. 1986. Beta-adrenergic receptor kinase: identification of a novel protein kinase that phosphorylates the agonist-occupied form of the receptor. Proc Natl Acad Sci U S A. 83:2797-2801.
Gilman, A.G. 1987. G proteins: transducers of receptor-generated signals. Annu Rev Biochem. 56:615-649.
Min, C., X. Zhang, M. Zheng, N. Sun, S. Acharya, X. Zhang, and K.M. Kim. 2017. Molecular Signature That Determines the Acute Tolerance of G Protein-Coupled Receptors. Biomol Ther (Seoul). 25:239-248.
Min, X., N. Sun, S. Wang, X. Zhang, and K.M. Kim. 2023. Sequestration of Gbetagamma by deubiquitinated arrestins into the nucleus as a novel desensitization mechanism of G protein-coupled receptors. Cell Commun Signal. 21:11.
Penela, P., A. Elorza, S. Sarnago, and F. Mayor, Jr. 2001. Beta-arrestin- and c-Src-dependent degradation of G-protein-coupled receptor kinase 2. EMBO J. 20:5129-5138.
Pitcher, J.A., Z.L. Fredericks, W.C. Stone, R.T. Premont, R.H. Stoffel, W.J. Koch, and R.J. Lefkowitz. 1996. Phosphatidylinositol 4,5-bisphosphate (PIP2)-enhanced G protein-coupled receptor kinase (GRK) activity. Location, structure, and regulation of the PIP2 binding site distinguishes the GRK subfamilies. J Biol Chem. 271:24907-24913.
Pitcher, J.A., J. Inglese, J.B. Higgins, J.L. Arriza, P.J. Casey, C. Kim, J.L. Benovic, M.M. Kwatra, M.G. Caron, and R.J. Lefkowitz. 1992. Role of beta gamma subunits of G proteins in targeting the beta-adrenergic receptor kinase to membrane-bound receptors. Science. 257:1264-1267.
Shenoy, S.K., and R.J. Lefkowitz. 2005. Receptor-specific ubiquitination of beta-arrestin directs assembly and targeting of seven-transmembrane receptor signalosomes. J Biol Chem. 280:15315-15324.
Touhara, K., J. Inglese, J.A. Pitcher, G. Shaw, and R.J. Lefkowitz. 1994. Binding of G protein beta gamma-subunits to pleckstrin homology domains. J Biol Chem. 269:10217-10220.
Zhang, X., X. Min, S. Wang, N. Sun, and K.M. Kim. 2020. Mdm2-mediated ubiquitination of beta-arrestin2 in the nucleus occurs in a Gbetagamma- and clathrin-dependent manner. Biochem Pharmacol. 178:114049.
Zheng, M., X. Zhang, X. Min, N. Sun, and K.M. Kim. 2020a. Cytoplasmic recruitment of Mdm2 as a common characteristic of G protein-coupled receptors that undergo desensitization. Biochem Biophys Res Commun. 530:181-188.
Zheng, M., X. Zhang, N. Sun, X. Min, S. Acharya, and K.M. Kim. 2020b. A novel molecular mechanism responsible for phosphorylation-independent desensitization of G protein-coupled receptors exemplified by the dopamine D3 receptor. Biochem Biophys Res Commun.
Round 2
Reviewer 2 Report
Comments and Suggestions for Authors
This reviewer appreciates and recognizes the effort of the authors in responding to the comments; however, the concerns raised in the first round of revision remain the same and were not addressed by the authors in this new version of the manuscript. Consequently, my opinion remains unchanged from the first review round, and I must recommend that this manuscript be rejected. The novelty of the study is limited, it lacks some crucial results and controls, has serious methodological flaws, and the statements and conclusions made by the authors are not fully supported by the data presented. The manuscript would benefit from a structural review and editing of the English language to be clearer.
Some specific comments and concerns are presented below but do not represent the entirety of the changes needed to consider this manuscript for publication:
IP negative controls should be included in all experiments based on immunoprecipitation to ensure that positive interactors labeling is specific for the IP protein.
Even though the focus of the work is mainly based on data obtained by immunoprecipitations, protein levels in total cell lysates are essential controls that must be included, even if only in supplementary figures.
Graphical presentation of the results would benefit if presented with individual data points instead of bar graphs to better show reproducibility and variability of the experiments.
The indications of the molecular weight marker presented in the final figures are not compatible if the images and labeling presented originate from the same membrane, as they should.
Description of results concerning spatial overlap quantification should be clearly included in the main text, plotted and included in the corresponding figures, and not merely referred to in the figure captions. Also, it's not clear how "the ratio of the space where Mdm2 is observed and the space surrounded by FLAG-D2R" was quantified.
Parameters related to the acquisition of microscopy images should be described in greater detail. It is not clear the total number of cells analyzed in the microscopy experiments and the data used for statistical analysis: 5 to 10 cells per experimental group in n=X independent experiments, or does the n=X represent the total number of cells analyzed?
Figure 1:
This concern was raised in the first round and answered by the autors: "Ensure that the labeling for the shown co-IP and IP, as well as the labeling for Lamin and β-actin, are from the same membrane."
Answer: They are obtained all from the same sample and the same membrane.
However, it's very clear, by the images provided in “supplementary files” and “original images” files, that the figure corresponding to the IP (labeled FLAG-Mdm2) and the co-IP (GRK2) do not originate from the same membrane as they should. Additionally, images presented for the lysate do not originate from the same membrane (Lamin and β-Actin) as they should. It is easily perceivable considering that in the FLAG-Mdm2 image there are 4 samples in a row, while GRK2 has 2 samples, followed by a molecular weight marker, followed by 2 samples. The same happens with Lamin/Actin images. Again, it's clear that the indications of the molecular weight marker presented in the final figures are not compatible with the claim that images originate from the same membrane.
Labelling of additional proteins would be important not only to show enrichment but also to demonstrate the exclusion of other cellular compartments like the ER.
Figure 2:
The indications of the molecular weight marker presented in the final figures are not compatible with the claim that images (ubiquitin and FLAG-GRK2) originate from the same membrane.
Samples from Con-KD cells should be included in the western blot showing the total levels of Arr2/3 (2B), only KD and overexpressing cells were included. Labeling in this figure indicating the proteins shown is absent. Interpretation of the data is easier when Co-IP and IP labeling are presented together and not intercalated with total lysate protein levels. Moreover, levels of FLAG-Mdm2 and Arr in total lysates should be presented for the same samples, to show the levels of re-expressed Arr in comparison with endogenous levels (Con-KD).
Figure 3:
Although it's clear to this reviewer the approach used, authors should clearly explain/expose how sequestration of Gβγ and why it was performed at the first use of this approach and not later in the manuscript.
Expression levels of GRK-CT should be presented.
The indications of the molecular weight marker presented in the final figures are not compatible with the claim that images (ubiquitin and FLAG-GRK2) originate from the same membrane.
The uncut membranes shown do not correspond to the versions presented in the manuscript.
Figure 4:
Although the authors refer to a previous publication, these data would be an important control to include, even if only in supplementary data.
Comments on the Quality of English LanguageThe manuscript would benefit from a structural review and editing of the English language to be clearer.
Author Response
This reviewer appreciates and recognizes the effort of the authors in responding to the comments; however, the concerns raised in the first round of revision remain the same and were not addressed by the authors in this new version of the manuscript. Consequently, my opinion remains unchanged from the first review round, and I must recommend that this manuscript be rejected. The novelty of the study is limited, it lacks some crucial results and controls, has serious methodological flaws, and the statements and conclusions made by the authors are not fully supported by the data presented. The manuscript would benefit from a structural review and editing of the English language to be clearer.
Some specific comments and concerns are presented below but do not represent the entirety of the changes needed to consider this manuscript for publication:
IP negative controls should be included in all experiments based on immunoprecipitation to ensure that positive interactors labeling is specific for the IP protein.
Even though the focus of the work is mainly based on data obtained by immunoprecipitations, protein levels in total cell lysates are essential controls that must be included, even if only in supplementary figures.
Graphical presentation of the results would benefit if presented with individual data points instead of bar graphs to better show reproducibility and variability of the experiments.
The indications of the molecular weight marker presented in the final figures are not compatible if the images and labeling presented originate from the same membrane, as they should.
Description of results concerning spatial overlap quantification should be clearly included in the main text, plotted and included in the corresponding figures, and not merely referred to in the figure captions. Also, it's not clear how "the ratio of the space where Mdm2 is observed and the space surrounded by FLAG-D2R" was quantified.
Ans: The circumference of the cell was measured using automated software (NIS-Elements AT program; Nikon Inc.)(Nikon Inc.)(Yoon et al., 2018).
Yoon, S., T. Kook, H.K. Min, D.H. Kwon, Y.K. Cho, M. Kim, S. Shin, H. Joung, S.H. Jeong, S. Lee, G. Kang, Y. Park, Y.S. Kim, Y. Ahn, J.R. McMullen, U. Gergs, J. Neumann, K.K. Kim, J. Kim, K.I. Nam, Y.K. Kim, H. Kook, and G.H. Eom. 2018. PP2A negatively regulates the hypertrophic response by dephosphorylating HDAC2 S394 in the heart. Exp Mol Med. 50:1-14.
Parameters related to the acquisition of microscopy images should be described in greater detail. It is not clear the total number of cells analyzed in the microscopy experiments and the data used for statistical analysis: 5 to 10 cells per experimental group in n=X independent experiments, or does the n=X represent the total number of cells analyzed?
Ans: The immunocytochemistry experiment was performed 2-3 times, and a total of 5 to 10 cells per experimental group were analyzed for each experiment. The experimental details in the revised manuscript are described more clearly.
Figure 1:
This concern was raised in the first round and answered by the autors: "Ensure that the labeling for the shown co-IP and IP, as well as the labeling for Lamin and β-actin, are from the same membrane."
Answer: They are obtained all from the same sample and the same membrane.
However, it's very clear, by the images provided in “supplementary files” and “original images” files, that the figure corresponding to the IP (labeled FLAG-Mdm2) and the co-IP (GRK2) do not originate from the same membrane as they should. Additionally, images presented for the lysate do not originate from the same membrane (Lamin and β-Actin) as they should. It is easily perceivable considering that in the FLAG-Mdm2 image there are 4 samples in a row, while GRK2 has 2 samples, followed by a molecular weight marker, followed by 2 samples. The same happens with Lamin/Actin images. Again, it's clear that the indications of the molecular weight marker presented in the final figures are not compatible with the claim that images originate from the same membrane.
Labeling of additional proteins would be important not only to show enrichment but also to demonstrate the exclusion of other cellular compartments like the ER.
Ans: I don't recall the details of each result because this project has been ongoing for over 5 years. Due to difficulties in clearly visualizing protein bands on a blot after multiple stripping and re-probing of the same membrane, the same sample was likely loaded 2-3 times to obtain blotting results for each protein. Following the referee's insistence on using results from the same membrane, we have updated the figures with data from a single membrane, even though these results might be less distinct than the previous ones.
Figure 2:
The indications of the molecular weight marker presented in the final figures are not compatible with the claim that images (ubiquitin and FLAG-GRK2) originate from the same membrane.
Samples from Con-KD cells should be included in the western blot showing the total levels of Arr2/3 (2B), only KD and overexpressing cells were included. Labeling in this figure indicating the proteins shown is absent. Interpretation of the data is easier when Co-IP and IP labeling are presented together and not intercalated with total lysate protein levels. Moreover, levels of FLAG-Mdm2 and Arr in total lysates should be presented for the same samples, to show the levels of re-expressed Arr in comparison with endogenous levels (Con-KD).
Ans: As requested by the referee, we replaced the figure with one showing results obtained from the same membrane. However, please note that the image quality of this result may be lower due to multiple stripping processes.
Figure 3:
Although it's clear to this reviewer the approach used, authors should clearly explain/expose how sequestration of Gβγ and why it was performed at the first use of this approach and not later in the manuscript.
Expression levels of GRK-CT should be presented.
The indications of the molecular weight marker presented in the final figures are not compatible with the claim that images (ubiquitin and FLAG-GRK2) originate from the same membrane.
The uncut membranes shown do not correspond to the versions presented in the manuscript.
Ans: As requested by the referee, we replaced the figure with one showing results obtained from the same membrane.
Figure 4:
Although the authors refer to a previous publication, these data would be an important control to include, even if only in supplementary data.
Ans: Based on the referee's suggestion, we tested the hypothesis that arrestin3 needs to enter the nucleus for ubiquitination using the dopamine D2 receptor. This finding is included in Fig.S1.
Reviewer 3 Report
Comments and Suggestions for Authors
The authors did a great job to clarify almost all questions I had. Few issues remain:
Q3-5. legend Fig. 1 Co-IP and IP: sounds as if there were two IPs performed, but as I understand one IP was done, with FLAG-beads, and blotted for FLAG and for GRK2. So just say Co-IP.
Ans: In this context, IP refers to the FLAG-tagged protein Mdm2, while Co-IP refers to the interacting protein, GRK2, which is immunoprecipitated along with Mdm2.
Yes, I get that, but for clarity I still think it is better to just say Co-IP, because it is one IP only.
Q3-11. Fig. 2A legend (also refers to other figures). Why do you indicate the concentration of D2R? First, the phrasing “transfected with D2R (1.7-2.1 pmol/mg protein) suggests you transfected protein, but I guess you transfected a plasmid coding for D2R. Second, what does this concentration tell, is it an indicator of overexpression, or similar expression to the endogenous? If not explained, these values are meaningless.
Ans: Cellular responses are largely dictated by receptor expression levels. Thus, when studying downstream events associated with receptors, it is essential to maintain comparable receptor expression levels across different experimental groups.
I´m not satisfied with the explanation. A) I insist that did not transfect protein but DNA. So you would have to say something like was transfected with D2R-coding plasmid resulting in D2R protein concentrations of 1.7-2.1 pmol/mg protein. How is that value measured? This can not be derived from the Western Blot, unless you have a recombinant D2R with known concentration for reference. Did you do quantitative mass spec?
Q3-18. l. 278. Why arrestin3 needs a partner with an NLS if it has one on its own?
Ans: For a protein to enter the nucleus through the nuclear pore complex, it must contain a nuclear localization signal (NLS). The protein gains entry by binding to the importin complex, which recognizes and a... to the NLS. This process is facilitated by Ran, a small G protein that uses GTP as its energy source.
I was aware of the general process of nuclear import. I was just wondering why arrestin3, which has an NLS, needs another protein with NLS. You wrote: “was suggested that arrestin3 enters the nucleus by binding with another protein that possesses an NLS”. I recommend some clarification, like “despite its own NLS, arrestin3 needs a binding partner with an additional NLS, or something like that
Author Response
The authors did a great job to clarify almost all questions I had. Few issues remain:
Q3-5. legend Fig. 1 Co-IP and IP: sounds as if there were two IPs performed, but as I understand one IP was done, with FLAG-beads, and blotted for FLAG and for GRK2. So just say Co-IP.
Ans: In this context, IP refers to the FLAG-tagged protein Mdm2, while Co-IP refers to the interacting protein, GRK2, which is immunoprecipitated along with Mdm2.
Yes, I get that, but for clarity I still think it is better to just say Co-IP, because it is one IP only.
Ans: We corrected the figure legend as the referee suggested.
Q3-11. Fig. 2A legend (also refers to other figures). Why do you indicate the concentration of D2R? First, the phrasing “transfected with D2R (1.7-2.1 pmol/mg protein) suggests you transfected protein, but I guess you transfected a plasmid coding for D2R. Second, what does this concentration tell, is it an indicator of overexpression, or similar expression to the endogenous? If not explained, these values are meaningless.
Ans: Cellular responses are largely dictated by receptor expression levels. Thus, when studying downstream events associated with receptors, it is essential to maintain comparable receptor expression levels across different experimental groups.
I´m not satisfied with the explanation. A) I insist that did not transfect protein but DNA. So you would have to say something like was transfected with D2R-coding plasmid resulting in D2R protein concentrations of 1.7-2.1 pmol/mg protein. How is that value measured? This can not be derived from the Western Blot, unless you have a recombinant D2R with known concentration for reference. Did you do quantitative mass spec?
Ans: I believe we may have misunderstood the referee's comment. Confirming receptor expression is such a routine process for us that we did not detail it in the manuscript. In the revised version, we have changed the figure legend to align with the referee's intent and described the radioligand binding process in the experimental methods section.
Q3-18. l. 278. Why arrestin3 needs a partner with an NLS if it has one on its own?
Ans: For a protein to enter the nucleus through the nuclear pore complex, it must contain a nuclear localization signal (NLS). The protein gains entry by binding to the importin complex, which recognizes and a... to the NLS. This process is facilitated by Ran, a small G protein that uses GTP as its energy source.
I was aware of the general process of nuclear import. I was just wondering why arrestin3, which has an NLS, needs another protein with NLS. You wrote: “was suggested that arrestin3 enters the nucleus by binding with another protein that possesses an NLS”. I recommend some clarification, like “despite its own NLS, arrestin3 needs a binding partner with an additional NLS, or something like that.
Ans: We apologize for not fully grasping the referee's concern initially. If the referee hadn't highlighted it again, this issue would have remained a significant flaw in the paper. To accurately convey our intent, the term "binds" should indeed be used instead of "possesses." However, upon revisiting this section, we realized it was overly concise and might be difficult for readers to understand. To prevent any confusion, we have decided to remove this part entirely. We appreciate the referee's thorough and insightful feedback.
Round 3
Reviewer 2 Report
Comments and Suggestions for Authors
This reviewer is satisfied with the authors' responses and the revisions made to the manuscript.
Comments on the Quality of English LanguageThe quality of the English language is generally good, but there are some sections where readability and flow could be improved.